# Migrating is not enough for modern planktonic foraminifera in a changing ocean

Sonia Chaabane[1,2,3 ✉], Thibault de Garidel-Thoron[1], Julie Meilland[4], Olivier Sulpis[1], Thomas B. Chalk[1], Geert-Jan A. Brummer[5], P. Graham Mortyn[6], Xavier Giraud[1], Hélène Howa[7], Nicolas Casajus[3], Azumi Kuroyanagi[8], Gregory Beaugrand[9] & Ralf Schiebel[2]

Rising carbon dioxide emissions are provoking ocean warming and acidification[1,2], altering plankton habitats and threatening calcifying organisms[3], such as the planktonic foraminifera (PF). Whether the PF can cope with these unprecedented rates of environmental change, through lateral migrations and vertical displacements, is unresolved. Here we show, using data collected over the course of a century as FORCIS[4] global census counts, that the PF are displaying evident poleward migratory behaviours, increasing their diversity at mid- to high latitudes and, for some species, descending in the water column. Overall foraminiferal abundances have decreased by $24.2 \pm 0.1\%$ over the past eight decades. Beyond lateral migrations[5], our study has uncovered intricate vertical migration patterns among foraminiferal species, presenting a nuanced understanding of their adaptive strategies. In the temperature and calcite saturation states projected for 2050 and 2100, low-latitude foraminiferal species will face physicochemical environments that surpass their current ecological tolerances. These species may replace higher-latitude species through poleward shifts, which would reduce low-latitude foraminiferal diversity. Our insights into the adaptation of foraminifera during the Anthropocene suggest that migration will not be enough to ensure survival. This underscores the urgent need for us to understand how the interplay of climate change, ocean acidification and other stressors will impact the survivability of large parts of the marine realm.

Ongoing anthropogenic carbon dioxide ($CO_2$) emissions are warming and acidifying the ocean[1,2], leading to water-column stratification[3] and altering ecological niches[6]. These effects are particularly severe for organisms producing a calcium carbonate shell or skeleton because acidification impedes calcification faster than warming favours it[7]. Furthermore, the increasing remineralization of organic matter in the upper water column[8,9], in response to ocean warming, could alter the availability of nutrients.

Ocean warming has already induced changes in planktonic habitats due to the inability of plankton to adapt fast enough to the increased physiological stress[5,10,11]. Comparable environmental crises have occurred in the geological past, but at much slower rates. For instance, significant surface ocean acidification has been reported from the last deglaciation and the onset of the Holocene[12]. Predictive models suggest that future warming and acidification will escalate, with negative ecological consequences for calcifying plankton[13–15] and with plankton communities shifting polewards[16,17]. However, the capacity of plankton to acclimate to ongoing (that is, on decadal timescales) changes, and to migrate in three dimensions, has been, until now, untested due to the previous lack of spatially and vertically resolved time series.

Among oceanic zooplankton groups, the PF are ubiquitous calcifying micro-organisms whose global distribution and fossil record make them an ideal model for bridging the geological and historical records of biodiversity. The presence and abundance patterns of PF are significantly influenced by temperature[18]. The displacement of PF over the last deglaciation (20–12 thousand years ago) showed a spatially heterogeneous pattern, highlighting the complex response of PF communities, and rendering the straightforward prediction of future changes difficult[19]. The adaptation of PF to the climate transition from the Last Glacial Maximum to the Holocene has been demonstrated as plausible through modelling and the fossil record. However, the current rate of change exceeds that of the last deglaciation by several orders of magnitude[20].

To investigate historical changes at the global scale, we made use of a census and synthesis of different biodiversity metrics for PF since the 1910s[4,21]. We aimed to elucidate the responses of PF communities to direct anthropogenic impacts by assessing interdecadal distribution patterns under ongoing environmental change. Specifically, we evaluated diversity changes and PF migrations in three dimensions. By exploring these migrations, together with environmental stressors, temperature and ocean acidity, we have provided a comprehensive

[1]Aix-Marseille Université, CNRS, IRD, INRAE, CEREGE, Aix-en-Provence, France. [2]Department of Climate Geochemistry, Max Planck Institute for Chemistry, Mainz, Germany. [3]Fondation pour la recherche sur la biodiversité (FRB-CESAB), Montpellier, France. [4]MARUM, Center for Marine Environmental Sciences, University of Bremen, Bremen, Germany. [5]NIOZ, Royal Netherlands Institute for Sea Research, Department of Ocean Systems, Texel, The Netherlands. [6]Universitat Autònoma de Barcelona, ICTA and Dept. of Geography, Barcelona, Spain. [7]LPG-BIAF, UMR-CNRS 6112, University of Angers, Angers, France. [8]Tohoku University Museum, Tohoku University, Sendai, Japan. [9]Université Littoral Côte d'Opale, Univ. Lille, CNRS, UMR 8187, Laboratoire d'Océanologie et de Géosciences (LOG), Wimereux, France. ✉e-mail: sonia.chaabane@gmail.com

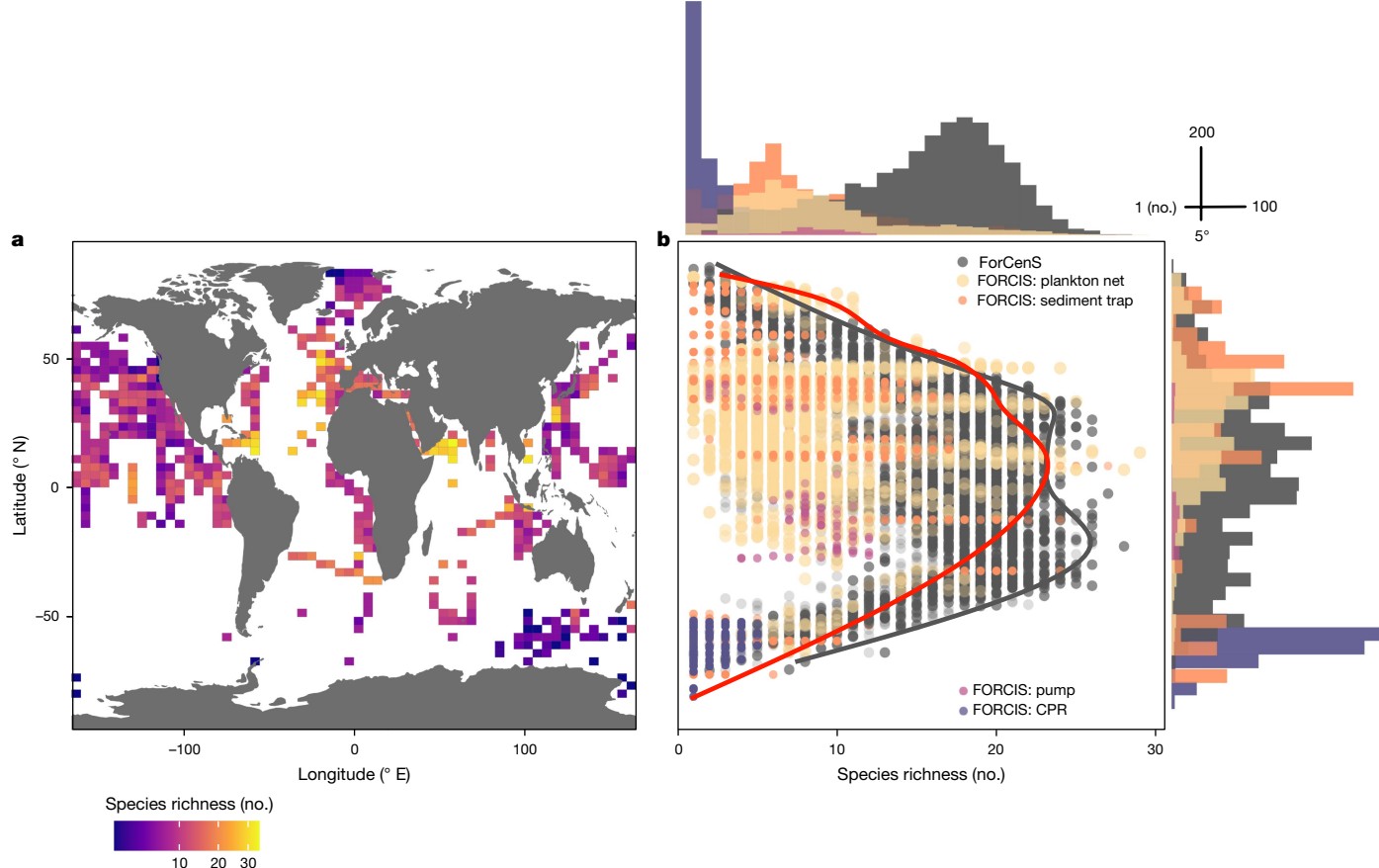

**Fig. 1 | Diversity changes in planktonic foraminifera. a**, Map showing diversity of PF (number of species based on the compiled taxonomy[4]). **b**, Comparison of pre-industrial diversity inferred from the surface-sediment ForCenS database with the living PF FORCIS database, based on collection using different sampling devices (plankton net, CPR, pump and sediment traps), for the past 100 years fitted using a generalized additive model smoothing curve at the 95th percentile of species richness for each 10° latitudinal bin (grey line, ForCenS data; red line, FORCIS data). The number of total observations from FORCIS and ForCenS together for each 5° latitudinal bin and each species richness level are presented in the histograms.

overview of PF adaptive strategies in a changing ocean. Furthermore, through simulations of future environments, we were able to discuss the potential trajectories and adaptive capacities of these species and, more broadly, of marine ecosystems.

## Latitudinal shift in diversity patterns

The effects of anthropogenic changes on plankton can be investigated through the analysis of long-term time series of observations[22,23]. The (Foraminifera Response to Climatic Stress (FORCIS) database comprises approximately 188,000 PF subsamples drawn from oceans across the globe since 1910[4,21]. Specifically, the regional data from the North Atlantic Ocean stands out for its remarkable temporal and spatial resolution, boasting PF time series that extend back, locally, over half a century (Extended Data Fig. 1).

The Atlantic mid-latitudes (30 to 50° N) are a key biodiversity hotspot, containing a wide spectrum of subtropical and temperate species (Fig. 1a), although the reported measured abundances are not particularly high (Extended Data Fig. 2). By comparing the species diversity of modern PF samples taken from the water column in the FORCIS database with the species diversity in PF samples from surface sediments (ForCenS database[23]), an evolution in the latitudinal diversity gradient during the early Anthropocene is apparent (Fig. 1b and Extended Data Fig. 3). Over the past century, the species diversity in each data profile from the mid- to high latitudes has slightly exceeded pre-industrial levels by four species, increasing from nine species in the pre-industrial data to 13 in the post-industrial data (between 65 and 80° N) (Fig. 1b).

In some regions in the North Atlantic and Pacific Oceans, the modern diversity surpasses pre-industrial levels by up to 10 species (Extended Data Fig. 3). Although both diversity estimates have been influenced by biases, such as preservation, ontogeny and seasonality, our comparison indicates that modern PF diversity is greater in high-latitude regions and lower near the Equator relative to pre-industrial levels.

Several PF species show a poleward shift between the pre-industrial (ForCenS database) and the modern (FORCIS) eras (Extended Data Fig. 4), implying a change between pre- and post-industrial distribution patterns. Notably, *Globorotalia scitula*, a sub-thermocline dweller[24–26] in the temperate ocean has expanded its ecological niche, according to FORCIS, now appearing at high latitudes up to 80° N (Extended Data Figs. 4 and 5). In addition, post-1990, several low-latitude species began to decline, potentially due to higher temperatures, leading to reduced diversity and a shift in the assemblage composition (Extended Data Fig. 6).

The post-industrial PF diversity decrease mirrors a decline in equatorial diversity in the pre-industrial era[27] and a diminished diversity in warmer low-latitude waters affecting PF distributions[28]. The latitudinal diversity shift observed from the FORCIS data is consistent with long-term changes in plankton assemblages, as predicted by modelling studies[20,29,30]. Such biogeographic shifts are in line with the spatial changes observed in various other marine and terrestrial species, seemingly influenced by increasing temperatures, particularly in the Northern Hemisphere[11,14,31–33]. Temperature may not only shape plankton distributions in the lower latitudes, but may also impact PF species distributions and diversity via its effect on the food supply[34].

## Tracking changes in ecological niches

The biodiversity loss observed at low latitudes appears to be unrelated to the trophic preferences of the PF groups, with PF species with or without photosymbionts (that is, symbiont-bearing and symbiont-barren), being spinose or non-spinose, tropical or subtropical all shifting towards higher-latitude habitats (Extended Data Fig. 7). Only two species (*Pulleniatina obliquiloculata* and *Globigerinoides ruber ruber*) have maintained a steady distribution pattern throughout the past century (Extended Data Fig. 7). Prominent species, such as *Globorotalia inflata, Globoturborotalita rubescens, Neogloboquadrina dutertrei* and *Globorotalia cultrata*, have migrated polewards, on average, at a pace of about 10.28 km yr$^{-1}$. Species such as *Globorotalia scitula, Trilobatus sacculifer* and *Globigerinoides ruber* display a poleward shift ranging between 4.11 and 10.28 km yr$^{-1}$. Likewise, marine ectotherms have been shown to migrate polewards at speeds of between 5.92 ± 0.94 km yr$^{-1}$ (ref. 35) and 7.20 ± 1.35 km yr$^{-1}$ (ref. 11), outpacing terrestrial species that have been migrating polewards at about 1.11 ± 0.96 km yr$^{-1}$ (ref. 35). This migration is consistent with recent modelling work[16], which has predicted a median poleward migration speed of 3.5 km yr$^{-1}$ for all plankton. The upper limit of the PF poleward migration speed (10.28 km yr$^{-1}$) corresponds exactly to the pace of thermal shifts in the oceanic isotherms (about 10 km yr$^{-1}$)[36]. This strongly suggests that PF closely track warming fronts, outpacing other planktonic groups.

A unique feature of the FORCIS database is the possibility to track vertical changes in PF distributions at the species level over time. At low latitudes, two species have significantly descended to greater depths over the past few decades. The symbiont-bearing *Globigerinella calida* exhibits a significant deepening of its habitat depth of 53.5 ± 15 m (from around 20 to 75 m). The symbiont-barren species *Globigerinita crassaformis* has descended by 45 ± 21 m (from around 55 to 100 m) between samples taken before and after 1997 (Fig. 2a). This notable shift in depth habitat may be attributed to changes in hydrological and ecological conditions, including changes in the water temperature or variations in food availability, which likely prompted their appearance in deeper waters. In the mid-latitudes, symbiont-barren species, such as *G. inflata* (40 ± 5 m, from around 30 to 70 m) and *Neogloboquadrina incompta* (40 ± 4 m, from around 10 to 50 m), have significantly descended in the thermocline and mixed layer (Fig. 2b). Only two out of eight symbiont-bearing PF species show significant vertical migration, that is, *Globigerinella siphonifera* (20 ± 5 m, from around 50 to 70 m) and *Orbulina universa* (40 ± 3 m, from around 10 to 50 m). Based on these observations, the trophic regime of PF species appears to have constrained the vertical distribution changes in species in the North Atlantic.

Regional and vertical shifts in PF distributions underscore how symbiont-bearing and symbiont-barren species respond to evolving environmental stressors. At mid-latitudes, tropical and subtropical symbiont-bearing species thrive in near-surface environments in the mixed layer (0–70 m), following the latitudinal shift of surface conditions towards oligotrophic waters. This has led to a shoaling of the majority of these species' habitats over our time series. The vertical descent of certain species whose habitats align with the deep chlorophyll maximum matches the anthropogenically driven deepening of the thermocline[6]. While specific species exhibit notable vertical migration, a comprehensive analysis of the PF in the FORCIS database indicated limited changes in the overall vertical distribution because most of the PF species do not show a deepening in their depth habitat. The lack of such deepening in some PF in low to mid-latitudes might be linked to remineralization processes nearer the ocean surface, as suggested by recent findings[8]. Elevated temperatures stimulate microorganism metabolism, expediting the degradation of organic matter. This process results in the availability of a more accessible food source near the ocean surface, potentially influencing the distribution patterns of specific PF species[8,9]. This suggests that vertical migration is influenced by the trophic regime of each species. However, this migration is slower than the deepening of the isotherms, which has been estimated at −6.6 ± 18.8 m decade$^{-1}$ between 1980 and 2015. This rate is predicted to accelerate to −32 m decade$^{-1}$ by the end of the century under a high emissions scenario[37]. We speculate this will limit the ability of PF to cope with these warmer environments.

## Planktonic foraminifera abundance decline

The latitudinal shift in diversity and the restricted vertical descent of the PF shows the limits of their ability to respond to environmental changes. However, a quantitative analysis of their abundances should indicate whether their phenotypic plasticity is broad enough to adapt to anthropogenic changes in one location. Our data revealed that the gradual decrease in surface and subsurface PF abundances across the different latitudinal bands in the North Atlantic Ocean over the past few decades was statistically significant between 0 and 50° N (Fig. 3). This abundance decrease is most pronounced in low- to mid-latitude regions (5.5 ± 0.05 (1 s.d.)% at 0 to 30° N and 24.24 ± 0.11% at 30 to 50° N between 1950 and 2018) (Fig. 3a,b and Extended Data Fig. 8), with the decline in abundance being particularly acute for subtropical and temperate species. These trends are even stronger when including the sparse data before 1950 from the low latitudes (0 to 30° N), with an abundance decline reaching 42.08 ± 0.15% from 1940 to 2018. Although early (pre-1960s) census data are rare, the post-1960s data show a statistically significant decline in abundance in 14 out of 26 species (for example, up to a maximum decline of 80 ± 0.3% between 1950 and 2010 for the subtropical species *G. siphonifera*). This general decline in PF abundance resembles the trends observed in other pelagic species[38]. In particular, zooplankton groups, such as copepods[30], have undergone notable biogeographic shifts over the past few decades, including a significant overall decline in the presence of colder-water species[30]. These findings indicate an overall habitat change in the marine plankton, especially in the context of PF abundance variations across latitudinal bands and time.

While the abundances of *Globigerinita glutinata* and *G. ruber ruber* have increased at low latitudes since 1940, an increase in *Globigerinita uvula* has occurred at high latitudes (Fig. 3a,c). By contrast, *T. sacculifer* has decreased in number at mid-latitudes (Fig. 3b). Two different, concurrent processes may explain this. First, a change in calcification intensity due to fluctuations in the carbonate chemistry could have provided an advantage to smaller species, such as *G. uvula* and *G. glutinata*[39]. Second, their expansion might be attributed to the wide temperature tolerance of *G. uvula*[40,41] (Extended Data Figs. 4 and 5). Overall, the response of PF species to anthropogenic changes shows a muted vertical habitat shift, differentiated by trophic regime, a latitudinal poleward shift tracking the rate of change in temperature fronts, and a general abundance decline reflecting their limited plasticity to evolving environments.

## Evolving ecological niches

The spatial distribution of PF is strongly controlled by temperature, with their maximum abundance generally matching some temperature optimum[18,28,42]. The observed sensitivity of PF to temperature has contributed to the establishment of a pronounced latitudinal diversity gradient[19,31,43], and has caused the discernible response to ongoing global warming reported here, as is also the case for other plankton groups[30,44]. The species-specific ecological niches of the marine biota are undergoing worldwide shifts due to warming and ocean acidification, partly resulting from anthropogenic $CO_2$ emissions[5,45]. We used the saturation state of seawater with respect to calcite ($\Omega_{calcite}$) as a metric for ocean acidification. Examining the prevalent PF species from three North Atlantic latitudinal-range bands, we found that their current habitats cover temperature and $\Omega_{calcite}$ ranges that largely align with, or exceed, projected mid-to-end-century changes under a realistic 'middle-of-the-road' shared socioeconomic pathway scenario (SSP2-4.5) (Extended Data Fig. 9). Notably, species such as *Globigerina bulloides* produce tests even in areas where the $\Omega_{calcite}$ is below 1, meaning that dissolution should be

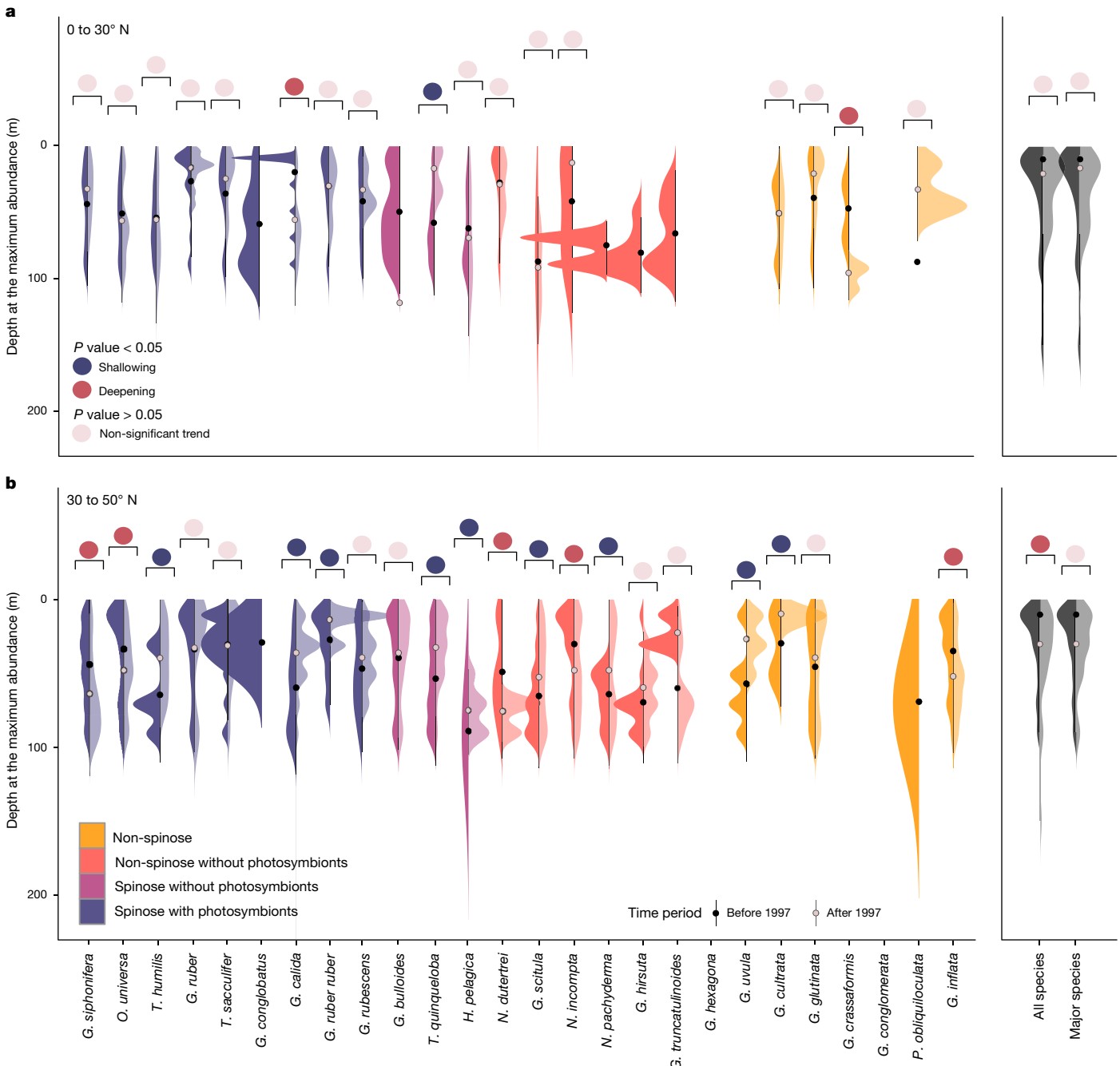

**Fig. 2 | Vertical distribution of planktonic foraminifera in the surface ocean. a,b**, Calculated depth of the maximum abundance before (dark-coloured half-violin plot) and after (light-coloured half-violin plot) 1997. The PF data are from samples from the North Atlantic Ocean, collected using plankton nets during spring and summer from between 0 and 200 m water depth, from 0 to 30° N (**a**) and 30 to 50° N (**b**). The black and grey dots show the mean water depths at the maximum abundance for before and after 1997, respectively, and the bars give the standard deviation. The *P* value obtained from the two-sided ANOVA test is reported as large coloured circles above each violin plot (light pink, non-significant trend; red, significant deepening; blue, significant shallowing).

favoured for calcite[46] (Extended Data Fig. 9). In the tropics (Extended Data Fig. 9a,d), by the end of this century, PF will be exposed to a combination of temperature and calcite saturation states that is unlike anything experienced by any other PF species today (Fig. 4). In temperate and high-latitude areas, PF are expected to reach the approximate limit of their currently habitable temperature and saturation state range by 2100. A broader analysis indicates that PF are relatively eurythermal, inhabiting a wide range of temperatures, but with species-specific temperature optima and distribution patterns (Extended Data Fig. 5).

By visualizing ecological niches in a two-dimensional ($\Omega_{calcite}$ and temperature) space, reflecting the environmental conditions, future migration patterns can be projected (Fig. 4). The distribution of $\Omega_{calcite}$ and temperature at the time and location of sampling for every PF occurrence in the FORCIS database provided the current PF ecological niches for three different latitudinal bins (Fig. 4). Each PF species occupies a subset of the current PF ecological niche, with some species adapted to warm, high-$\Omega_{calcite}$ waters (for example, *G. ruber*) and some being more familiar with cold, low-$\Omega_{calcite}$ waters (for example, *Neogloboquadrina pachyderma*). Using model predictions of future temperature and $\Omega_{calcite}$, under the SSP2-4.5 scenario, the future trajectory of these regional niches can be predicted, in terms of temperature and $\Omega_{calcite}$. Contrary to common perception, it is tropical, and not polar,

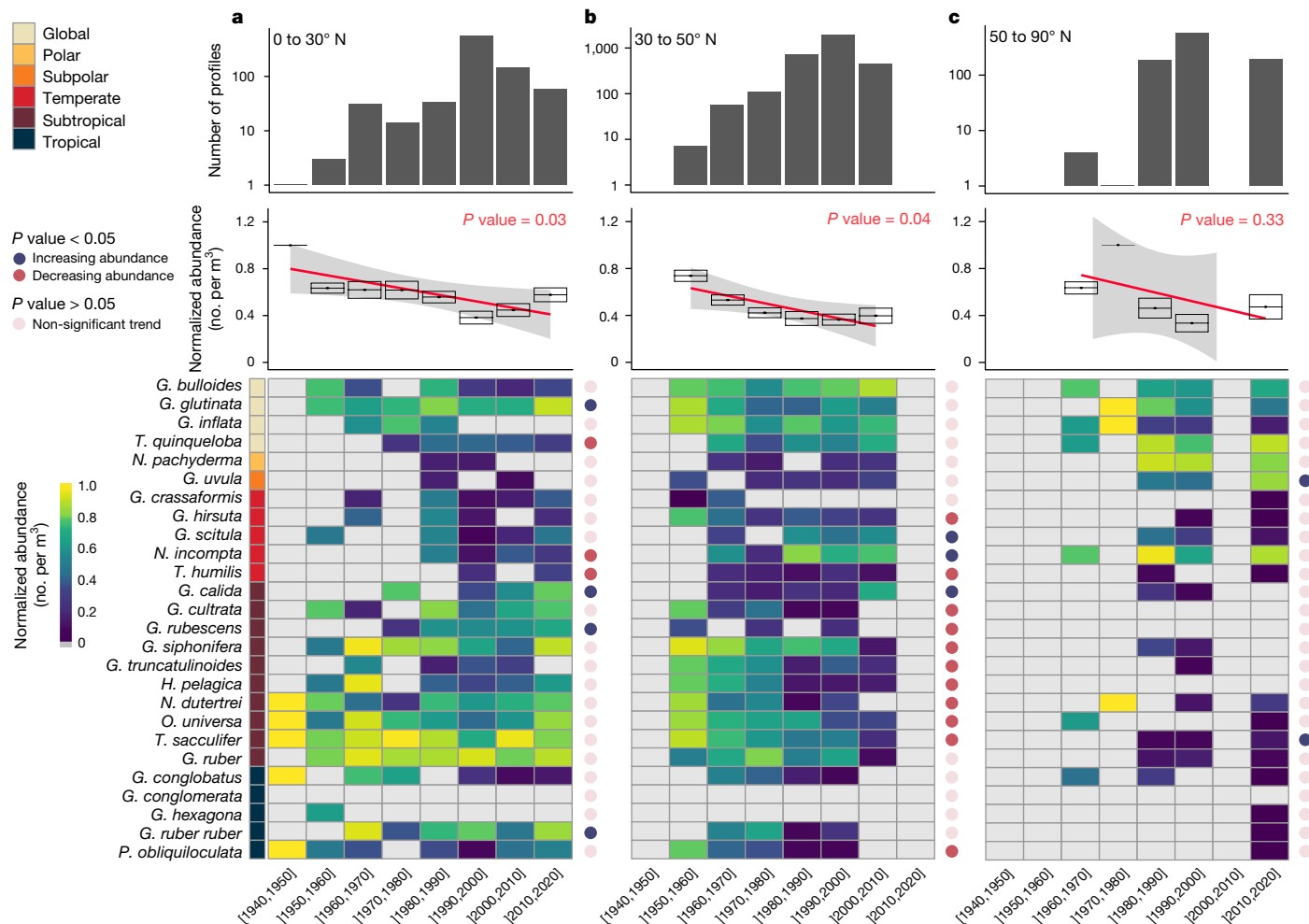

**Fig. 3 | Temporal variations in planktonic foraminifera abundance.**
**a–c**, Heatmaps (bottom panels) showing the normalized abundances of the different modern PF species in FORCIS, which were collected from 0 to 200 m depth over the past several decades, using a plankton net and a plankton pump, from the Arctic and North Atlantic Ocean latitudinal bands 0 to 30° N (**a**), 30 to 50° N (**b**) and 50 to 90° N (**c**). The P value obtained from the two-sided ANOVA is reported as coloured circles for each species per latitudinal band (light pink,

non-significant trend; red, significant decrease; blue, significant increase). The upper panels display the number of observations for each decade and latitudinal band. The regression lines (red) in the middle panels represent the relationship between the mean values of the normalized abundances and the decades, with error bars representing ±1 σ derived and the plots shown as minimum/maximum, with the median being the horizontal line.

environments that will probably see the most important changes, despite the polar amplification of climate change. By 2050, the environmental conditions in most tropical locations will fall outside those of the current PF ecological niches. In temperate and polar zones, the current PF niches will also move towards the limits of modern communities, although they will largely remain within the temperature and $\Omega_{calcite}$ conditions currently occupied globally by PF. It is probable that conditions such as these, with higher sea-surface temperatures and lower $\Omega_{calcite}$ than currently anywhere present on Earth today, have occurred in the geological past, but with different PF communities. Our knowledge about past analogues is still limited, but (less-)rapid warming has driven all the PF genera to extinction in the past (for example, ref. 47). By 2050, assuming non-adaptive responses in PF species, a significant latitudinal migration is projected, from low to mid- to high latitudes, with no compensatory PF species replacements.

Our observations are in agreement with the results of modelling studies[15,20]. For instance, Roy et al.[15], using the FORAMCLIM model for future simulations, showed that PF abundance and diversity are projected to decrease in the tropics and subpolar regions and increase in the subtropics and polar regions. It is unlikely that warming and ocean acidification in the mid- and high latitudes, where key PF species dwell in low-temperature and high-$\Omega_{calcite}$ waters, will affect the current PF

ecological niches by 2100. It is more possible that, by 2100, the mid- and high-latitude regions will switch from the current species niches to others (Fig. 4a,b), with current PF communities being replaced by other PF species adapted to warmer environments, as has already been observed in poleward migrations over the past few decades.

Such a dramatic extirpation of PF species from the tropics under the predicted, and already observed, decline in $\Omega_{calcite}$, particularly evident at high latitudes[48], does not necessarily imply the total absence of those species. Ultimately, this could lower calcification rates or limit calcification entirely, leading to the emergence of shell-less PF. A culture experiment by Evans and Erez[49] confirmed that two PF species, *G. ruber* and *G. siphonifera,* can survive post-shell dissolution, recalcify and adapt to low-pH conditions. The recalcification of dissolved foraminiferal tests has been validated under both field and laboratory conditions[50–52], suggesting that PF could live shell-less in low-$\Omega_{calcite}$ regions as predicted by future climatic scenarios (Fig. 4). However, it remains unclear whether they can reproduce under such conditions. Reduced calcification would marginally affect the calcite cycle because about 0.1–3.8% of the global surface carbonate flux in the modern open ocean originates from PF (ref. 53). Although past large-scale acidification events in Earth's history (for example, at the Paleocene–Eocene Thermal Maximum[54]) have not shown significantly

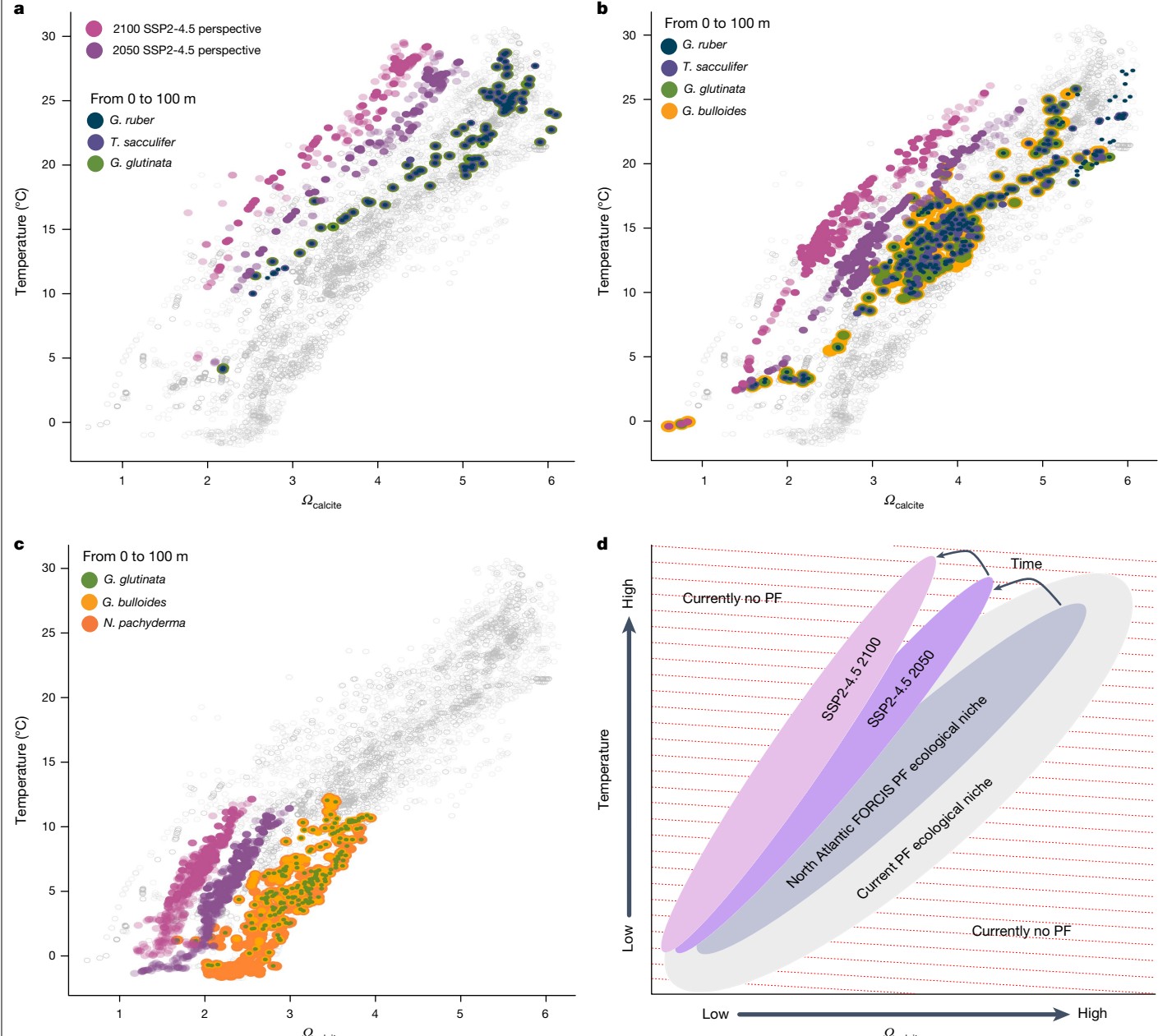

**Fig. 4 | Projected changes in ocean temperature and calcite saturation in foraminifera habitats by 2050 and 2100. a–d,** Modelled ocean temperature and $\Omega_{calcite}$ at collection time and collection water depth from the IPSL-CM6A-LR model output for all FORCIS foraminifera locations globally (grey points). Marker species are shown as coloured points with *G. ruber* (blue), *T. sacculifer* (dark purple) and *G. glutinata* (green) in the tropics (**a**), *G. ruber* (blue), *T. sacculifer* (dark purple), *G. glutinata* (green) and *G. bulloides* (yellow) at mid-latitudes (**b**) and *G. glutinata* (green), *G. bulloides* (yellow) and *N. pachyderma* (orange) at high latitudes (**c**). Also shown in each panel are the projected future conditions of all current niches of the PF marker species for each latitudinal band in 2050

(purple) and 2100 (pink) based on the model output SSP2-4.5. The pink and purple distributions show that future oceanic environments will be under temperature and saturation conditions that are not currently experienced by any living PF species. **d,** Cartoon schematic of ocean temperatures and saturation states in the niches currently inhabited by PF. The grey balloons cover the actual conditions. The purple balloon represents the historical PF localities and their conditions projected to 2050. The pink balloon shows these locations projected to 2100. The only Atlantic group outside of known modern conditions is the tropical biota, which will (if it remains tropical) face unprecedented combinations of temperature and saturation states.

altered long-term calcite flux to the seafloor, the exceptional magnitude and pace of the current acidification, combined with other potential stressors, suggest that PF are entering an uncharted era of environmental change, and that a related negative global-scale impact on calcification is plausible[20].

The consequences of acidification may affect organisms at various levels across the food chain. Photosynthesizing calcareous nannoplankton (for example, the coccolithophores) have shown marked patterns of decreased calcification upon acidification[55]. For other groups, such

as picoplankton (for example, cyanobacteria and small eukaryotes), echinoderms, crustaceans and cephalopods, the calcification response is more nuanced and is mediated by phenotypic plasticity[56,57]. Nevertheless, for most calcifying groups, as well as certain fish groups that produce highly soluble forms of calcium carbonate[58], acidification effects may be particularly detrimental in their early-life stages[56,59]. The calcifying community's response will be further complicated by the conflicting influences of changing nutrients, temperature and other environmental parameters.

Our study shows that modern PF are unlikely to keep pace, on decadal timescales, with current anthropogenic environmental changes, and will respond largely through declining abundances and latitudinal migration. Notably, their vertical descent has been limited in the low to mid-latitudes. Projections of future environmental change on modern PF niches have forecast their potential near-disappearance from low latitudes, raising the question of their adaptability to unprecedented change. For modern PF, migration is not enough.

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

# Methods

## Data

The PF data were extracted from the FORCIS[4,21] and ForCenS[23] databases and covered the post-industrial and pre-industrial time periods, respectively. Seasons for both databases were distinguished between the Northern and Southern Hemispheres and followed the meteorological seasons. For the Northern Hemisphere, autumn was defined as September, October and November, winter as December, January and February, spring as March, April and May, and summer as June, July and August. For the Southern Hemisphere, spring was defined as September, October and November, summer as December, January and February, autumn as March, April and May, and winter as June, July and August.

**FORCIS database.** More than 188,000 subsamples (each comprising one single plankton aliquot collected within a certain depth range, time interval and size-fraction range, at a single location), taken from around 163,000 samples collected from different oceanographic environments using plankton nets (approximately 22,000 subsamples from about 6,000 samples), continuous plankton recorders (CPRs) (approximately 157,000 subsamples), sediment traps (approximately 9,000 subsamples) and plankton pumps (approximately 400 subsamples) from the global ocean (Extended Data Fig. 1), were extracted from the FORCIS database[4,21]. These data included published and unpublished post-industrial data from the literature between 1910 and 2018. The total abundances of the PF, and species-specific counts for these samples, were extracted from FORCIS.

**ForCenS database.** Around 5,000 samples, covering the pre-industrial period, form the basis of the ForCenS database[23] of PF census counts from marine surface-sediment samples. This database only includes PF tests larger than 150 µm in diameter. In our study, the species abundances were normalized to 0 and 1 to represent species absence and presence, respectively.

## Data and taxonomic harmonization

To perform data harmonization and address methodological and taxonomic biases, we converted the abundance data extracted from FORCIS into abundance (individuals per m³). Specifically, where coarse fractions were sampled using a mesh size greater than 100 µm, we employed the approach described in Chaabane et al.[60] for standardization. This approach involves converting the abundance data extracted from different sampling devices, such as plankton nets and pumps, into a common unit of individuals per cubic metre (individuals per m³). To achieve this, we used size-normalized catch model equations, obtained from the sampling depths, on the total PF for cytoplasm-filled and empty tests. We were able to accurately quantify the abundances in the coarse fractions down to 100 µm, by applying the following equations:

$$C^{\infty}_{\mathrm{sz\_norm}} = C^{\mathrm{sz\_sup}}_{\mathrm{sz\_inf}} \frac{f_{\max} - f_{\mathrm{sz\_norm}}}{f_{\mathrm{sz\_sup}} - f_{\mathrm{sz\_inf}}}$$

where sz_inf and sz_sup denote the lower and upper size limits of the measured size fraction, respectively, sz_norm represents the normalization size, and $f_{\mathrm{sz}}$ is the multiplication factor associated with sz, calculated as follows:

$$f_{\mathrm{sz}} = 1 + (f_{\max} - 1) \frac{(S_{\mathrm{sup},k} - S_{\mathrm{sup},1})}{(S_{\mathrm{sup},k} - S_{\mathrm{sup},1}) + (S_{\mathrm{half}} - S_{\mathrm{sup},1})}$$

with $S_{\mathrm{sup},1}$ and $S_{\mathrm{sup},k}$ being the upper size limits of size classes 1 and $k$, respectively, while $S_{\mathrm{half}}$ and $f_{\max}$ are reported in ref. 60.

This facilitated computation of the assemblages as they would have appeared if all the material had been sampled using a 100 µm net, thus ensuring consistency across different sampling devices and conditions. Predicting the abundances of very small and rare species is still challenging, and so those data are not interpreted here.

The FORCIS dataset has been assessed and corrected for taxonomic discrepancies over the past century[4], and although some species might have been misidentified, this appears relatively unlikely because, overall, mostly large PF, whose taxonomic determination is easier, were collected in the early part of the record (that is, due to large net mesh sizes in the 1960s). All species counts were generated for both the 'lumped' and 'validated' taxonomies proposed by the FORCIS group[4] and based on the analytical reasons. For this study, several morphospecies were grouped (lumped) together for data analysis, such as *G. ruber* (*Globigerinoides ruber albus* + *Globigerinoides elongatus*), *Globorotalia truncatulinoides* (*G. truncatulinoides* left + *G. truncatulinoides* right) and *T. sacculifer* (*T. sacculifer* no sac + *T. sacculifer* with sac). However, *G. ruber ruber* (pink) was analysed separately from *G. ruber* (white). Based on their species-specific habitat preferences, the PF assemblages were grouped into six provinces (Supplementary Table 1), that is, tropical, subtropical, temperate, subpolar, polar and global[34]. Additionally, they were split into three groups based on their food preference regime, that is, symbiont-bearing, symbiont-barren and facultatively symbiont-bearing, following the work of Takagi et al.[61] and Hemleben et al.[62].

## Data analyses

To account for sampling biases, the following corrections were applied: (1) some analyses were focused on latitudinal bands to dilute the effect of under-sampled regions through time. The latitudinal bands were selected based on the assemblages' provinces and on Bé and Tolderlund[63] and Schiebel and Hemleben[34]; (2) the sample depth selection was limited for most of the analyses to the depth where most of the individuals were sampled alive so as to not confuse between depth habitat and post-mortem assemblages; (3) most of the analyses were limited to the spring and summer blooms in the North Atlantic Ocean, where the sampling coverage and number of samples were highest (Extended Data Fig. 1); (4) the abundances were standardized from 0 to 1 to correct for the number of samples and to enhance their visualization in heatmaps; and (5) to analyse the first and second halves of the dataset, distinct cut-off points were established. To separate the dataset in time, into two fractions of equal size, the cut-off date was set at 1990. In Fig. 2, the cut-off year is set at 1997 because this figure is based solely on multinet data, which have only been available since 1989. This allowed the comparison of two datasets with similar amounts of information.

**Environmental data.** Temperature and $\Omega_{\mathrm{calcite}}$ estimates associated with each PF from the North Atlantic samples (Fig. 4 and Extended Data Fig. 9) were taken from the Institut Pierre-Simon Laplace global climate model IPSL-CM6A-LR (ref. 64), in its Coupled Model Intercomparison Project Phase 6 (CMIP6) historical simulation, version r22i1p1f1, with a monthly resolution. The model output was converted to the World Ocean Atlas spatial grid, with a 1 × 1° resolution. The $\Omega_{\mathrm{calcite}}$ was computed by dividing the modelled in situ carbonate-ion concentration (denoted 'co3' in the CMIP6 nomenclature) by the carbonate-ion concentration at saturation with respect to calcite (denoted 'co3satcalc'). The temperature and $\Omega_{\mathrm{calcite}}$ were then associated with each sample in the FORCIS database by linearly interpolating the model output temperature to the latitude, longitude, water depth and time of sampling. For the multinet, CPR and pump data, we used the averaged sampled depth. For the time series (Fig. 4a–f and Extended Data Fig. 9), we averaged the data annually, and over the latitudinal ranges in the Atlantic Ocean, but only considering those values that corresponded to locations where a PF sample was present. We extracted the locations and water depths corresponding to each PF sample from the IPSL-CM6A-LR SSP2-4.5 simulation, which corresponds to future

predictions in response to the atmospheric $CO_2$ growth trajectories of the SSP2-4.5 middle-of-the-road scenario, to calculate how temperature and $\Omega_{calcite}$ would change throughout the 21st century.

**Biodiversity change.** The species richness was calculated for each 3° latitude × 6° longitude box and for time for each data point from the plankton net, pump and CPR profile and sediment trap sample in FORCIS (Fig. 1a and Supplementary Figs. 1A and 2). To address the potential disparity between the sediment trap and plankton net data, the sediment trap samples, which were collected over an average period of 15 days, were compared to the plankton net profiles that consisted of multiple samples gathered from the same location and within the same time interval. The sediment trap collection periods captured a representative range of species comparable to those found in the plankton net profiles taken from similar locations (Supplementary Fig. 3). In addition, the latitudinal diversity gradient for the ForCenS data was determined by calculating the number of species present in each sample and at each latitude, and comparing these to those obtained from the FORCIS plankton net, pump and CPR profiles and sediment trap samples (Fig. 1b and Supplementary Fig. 1B). Then, for each 10° latitudinal bin, the 95th percentile of species richness was selected, assuming discrete sampling using nets, CPRs and pumps. These selected data points were then fitted with a generalized additive model to capture the variability in species richness along the latitudinal gradient. These analyses focused on 26 species that were present in both the ForCenS and FORCIS databases, using the same taxonomic criteria. Species that were absent from ForCenS, such as *Bolivina variabilis, Tenuitellita fleisheri* and *Tenuitellita parkerae*, were not considered because comparison between the datasets would not be possible. In a second step, our assessment aimed to determine any species loss or gain from the pre-industrial to the post-industrial time periods. For this analysis, the diversities in FORCIS and ForCenS were evaluated for each 4.5° latitude × 9° longitude box, considering data from the plankton net, pump and CPR profiles and sediment trap samples in FORCIS and each sample in ForCenS. The species richness from FORCIS was subtracted from that from ForCenS within each grid to identify any changes (losses or gains) in species richness. This analysis focused on the major species (26 species) on a presence/absence basis (Extended Data Fig. 3). To contrast the distribution patterns between the post-industrial (FORCIS) and pre-industrial (ForCenS) samples, the species counts were transformed into presence and absence data, ensuring uniform taxonomy across both databases. The data were visualized on a grid map, with each grid cell representing 2.8° of latitude and 5.6° of longitude. From the 26 species under consideration, nine were specifically chosen based on their main environmental niches (including polar and subpolar, tropical to subtropical, globally distributed and deep-sea species), significant relevance to palaeoceanography and the introduction of new insights (for example, *G. uvula*) (Extended Data Fig. 4).

**Spatial and vertical migration of planktonic foraminifera.** The species' poleward migration was first assessed from the FORCIS database. Thus, the loss and gain of species in the northernmost limit of the North Atlantic and Arctic Oceans was assessed on the PF living at depth ranges of between 0 and 100 m during spring and summer before and after the cut-off year of 1990. The maximum latitude of the northernmost 5% of samples was calculated for all species before and after 1990. The difference between the maximum latitude before and after 1990 was evaluated to assess the direction and magnitude of the latitudinal migration of the different species (Extended Data Fig. 7). The selection of various cut-offs for spatial migration was influenced by the quantity of data available, both before and after applying specific filters, which were sometimes very few between 1910 and 1970.

The analyses were complemented by assessing the vertical occurrences of the PF through the water column. The vertical ranges of the PF were assessed for two latitudinal bands—0 to 30° N and 30 to

50° N—in the North Atlantic Ocean (Fig. 2). Only multinet data sampled across the upper 200 m and profiles with four or more samples and the same sampling resolution (for example, depth separations) during both spring and summer were selected. These data cover the period between 1980 and 2018. We focused on spring and summer due to the increased biological productivity at this time, the warmer sea-surface temperatures, and the greater data availability from the more-frequent research cruises occurring during these seasons. The depth of maximum abundance was calculated for each profile and for each species, and then the results before and after the cut-off year (1997) were compared (Fig. 2). To assess whether there were significant changes in depth associated with the maximum abundances of the different species in each latitudinal band and through time, an analysis of variance (ANOVA) was conducted (Fig. 2 and Supplementary Table 2).

**Abundance changes. Changes through time.** To determine the temporal variation in PF abundance, we analysed the PF per cubic metre (no. per m³) collected using plankton nets and pumps across different depths (0 to 200 m) and geographical regions (from the North Atlantic to the Arctic Oceans). These data were aggregated into three distinct latitudinal bands: 0 to 30° N, 30 to 50° N and 50 to 90° N (Fig. 3). For each decade since 1940, we normalized the total abundance of PF for each species within these bands. Normalization from 0 to 1 was performed to facilitate comparison across species and regions by standardizing the data, removing the effects of differing scales of abundance. This approach allowed us to observe relative changes in abundance over time, making it easier to identify trends and patterns. To investigate whether there were significant changes in the abundances of different species in each latitudinal band, an ANOVA was conducted using the anova() function on the fitted regression models. The ANOVA helped to determine whether the observed variations in species abundance across time bins were statistically significant (Fig. 3 and Supplementary Table 3). The ANOVA calculated a *P* value that indicated the probability of observing the differences in abundance between species by chance alone. A low *P* value (below 0.05) suggested that the observed differences were unlikely to be due to random variation and more likely to represent real change. Then, similarly to the previous analysis, we assessed the abundances of the different species collected using plankton nets and pumps only, and over a larger depth range of between 0 and 300 m, from the North Atlantic to Arctic Oceans, and in each latitudinal band with a resolution from 20 to 40° N and for each species (Extended Data Fig. 8).

**Changes with latitude.** For the data presented in Extended Data Fig. 2, only those studies that provided species counts for the Atlantic, Antarctic and/or Arctic Oceans, using plankton net data sourced from the FORCIS database, were considered because these presented the best temporal and spatial data coverage for statistically significant analysis. The selected studies covered the time period from 1980 to 2018 (Extended Data Fig. 2). The depth profiles of these data ranged from 0 to more than 100 m water depth. To examine the latitudinal abundance trend of PF species over time, the surface data (down to 100 m) and deep-water samples (below 100 m) from 1980 onwards were plotted against latitudes grouped in 10° intervals. This analysis specifically focused on the spring and summer seasons, which had the best-documented data.

**Thermal habitat changes.** To study the thermal preferences of the PF species in the global ocean, the abundances obtained from the plankton net, pump and CPR samples from above 100 m water depth were normalized from 0 to 1, and each species was assessed in each 1 °C binned and extracted in situ temperature from the Reanalysis Data Hadley EN 4.2.1 analyses g10 at 1 × 1° resolution, and assessed for the different time periods, before and after 1990 (Extended Data Fig. 5).

To test the effect of high temperatures on the PF assemblages at low latitudes (30° S to 30° N), the normalized abundances of the FORCIS samples, collected from data from net-tows during spring and summer

over a depth range of 0 to 100 m and before and after the cut-off date (1990) ($n = 1,000$) (Extended Data Fig. 6), were plotted against their binned (at each 1 °C) in situ temperatures, between 15 and 32 °C, and sourced from the Hadley EN 4.2.1 analyses g10 at a 1 × 1° resolution. The mean values of the normalized abundances and standard errors between samples were assessed and plotted against the binned temperature to assess their evolution with increasing temperature. The number of profiles was assessed for each 1 °C temperature bin and per species (Supplementary Table 4). Cluster analyses were applied to the data, based on principal coordinate analyses before the Euclidean distance computation between the species, to cluster the species assemblages that had the same response. The Euclidean distance matrix was then computed from the scores of the two first principal components, with those consistently explaining more than 86% of the total variance (dim1 = 79.7%, dim2 = 6.9% before cut-off; dim1 = 66.6%, dim2 = 21.8%) (Extended Data Fig. 6).

### R packages

The computational analysis conducted in this study utilized a variety of open-source tools in the R environment v.4.1.2 (ref. 65). In the comprehensive analysis of PF abundance variations, multiple high-performance R packages were deployed. For handling string manipulations and pattern matching, stringr was utilized[66]. dplyr allowed for robust data transformation and filtering[67], while vegan was used to conduct the ecological multivariate data analyses[68]. Reading and writing the Excel files was made seamless using openxlsx[69], whereas the phylogenetic and evolutionary analysis was facilitated by ape[70]. The package pheatmap allowed for the creation of heatmaps[71]. Clustering and partitioning of the data to identify patterns were executed using cluster[72], while the multivariate data analysis results were extracted and visualized using factoextra[73]. The viridis package supplied colourblind-friendly colour palettes[74], and tidyr enabled easier data cleaning and wrangling[75]. ggplot2 and ggpubr were used to create high-quality graphics[76,77], with reshape2 and reshape facilitating the reshaping of the data structures[78]. The plotrix package provided additional plotting functionalities[79]. Performance and risk calculations were executed using PerformanceAnalytics[80], while the correlation matrices were visualized using ggcorrplot[81]. For visualization scaling, scales was applied[82]. Spatial data visualization was carried out using ggspatial and ggmap[83,84], and the geographical maps were drawn using maps[85]. The ggExtra package enriched the ggplot2 graphics[86]. Lastly, the gridExtra package enabled the arrangement of multiple grid-based plots[87]. This extensive usage of high-performance R packages significantly contributed to the robust, reproducible and efficient data analysis in this work.

### Reporting summary

Further information on research design is available in the Nature Portfolio Reporting Summary linked to this article.

## Data availability

The FORCIS database used for this paper is available at Zenodo (https://zenodo.org/record/8186736)[88]. ForCenS database is also available from https://doi.pangaea.de/10.1594/PANGAEA.873570.

## Code availability

Codes to normalize the abundance data are available at Zenodo (https://doi.org/10.5281/zenodo.10750545)[89]. All code used for data analysis and generation of figures related to this Article is available at Zenodo (https://zenodo.org/records/10881387)[90].

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

**Acknowledgements** The FORCIS project is supported by the French Foundation for Biodiversity Research (FRB) (https://www.fondationbiodiversite.fr/) in the Centre for the Synthesis and Analysis of Biodiversity (CESAB) (https://www.fondationbiodiversite.fr/la-fondation/le-cesab/) and co-funded by the INSU LEFE programme, and the Max Planck Institute for Chemistry (MPIC) in Mainz, Germany. This work received support from the French government under the France 2030 investment plan, as part of the Initiative d'Excellence d'Aix-Marseille Université (A*MIDEX AMX-20-TRA-029).

**Author contributions** The study was designed by S.C., T.G., R.S., J.M., O.S., T.C., G.B. and G.M. during the FORCIS project workshops at FRB-CESAB. All authors contributed to the interpretation and discussion of the results. S.C. carried out the data analysis and wrote the paper, with contributions from T.G., R.S., J.M., O.S., T.C., G.B., G.M., X.G., H.H., N.C., A.K. and G.B. Figures and statistical analyses were generated by S.C., O.S. and T.C.

**Funding** Open access funding provided by Max Planck Society.

**Competing interests** The authors declare no competing interests.

**Additional information**
**Correspondence and requests for materials** should be addressed to Sonia Chaabane.

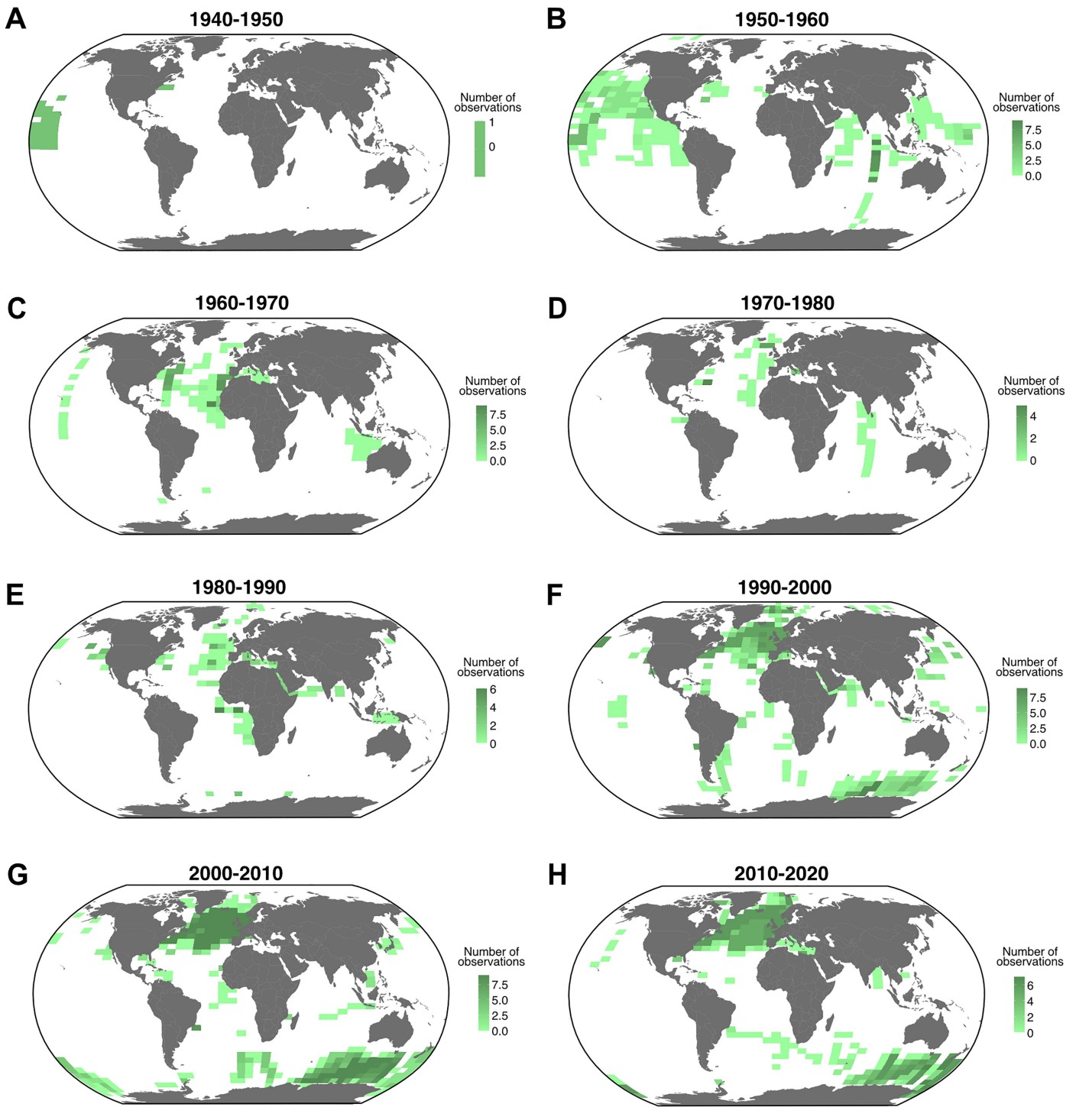

**Extended Data Fig. 1 | Temporal Coverage of FORCIS Database per Decade.** Geographical distribution of the data collected for each decade, in the FORCIS database from 1940 to 1950 (**A**), from 1950 to 1960 (**B**), from 1960 to 1970 (**C**), from 1970 to 1980 (**D**), from 1980 to 1990 (**E**), from 1990 to 2000 (**F**), from 2000 to 2010 (**G**), and from 2010 to 2020 (**H**).

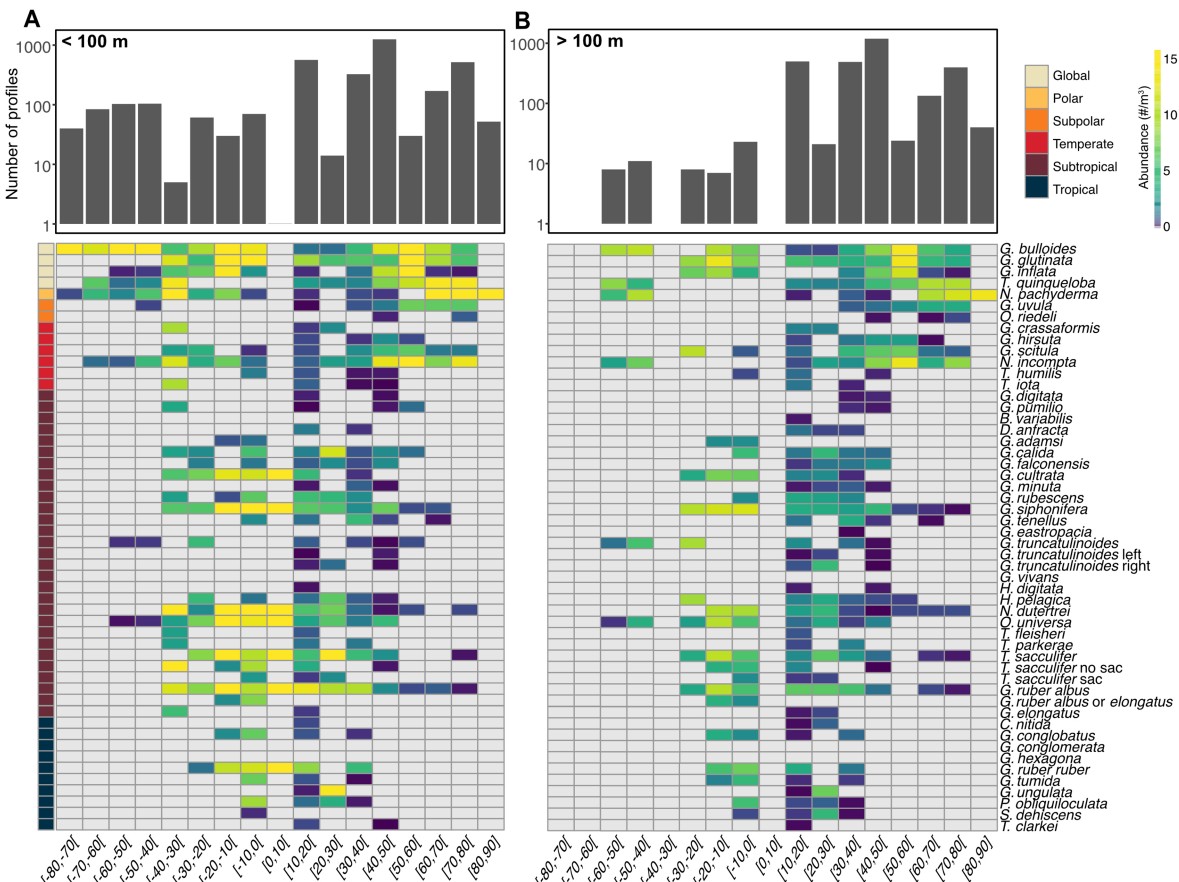

**Extended Data Fig. 2 | Latitudinal Variation in Planktonic Foraminifera Abundance.** Abundance variations of the different planktonic Foraminifera species in FORCIS collected using plankton net from the Arctic, Antarctic, and Atlantic oceans after 1980 during summer and spring and across latitudinal bins of 10°. (**A**) On the surface to 100 m, and (**B**) below 100 m. The number of observations at each 10° latitudinal-band is given in the upper panels.

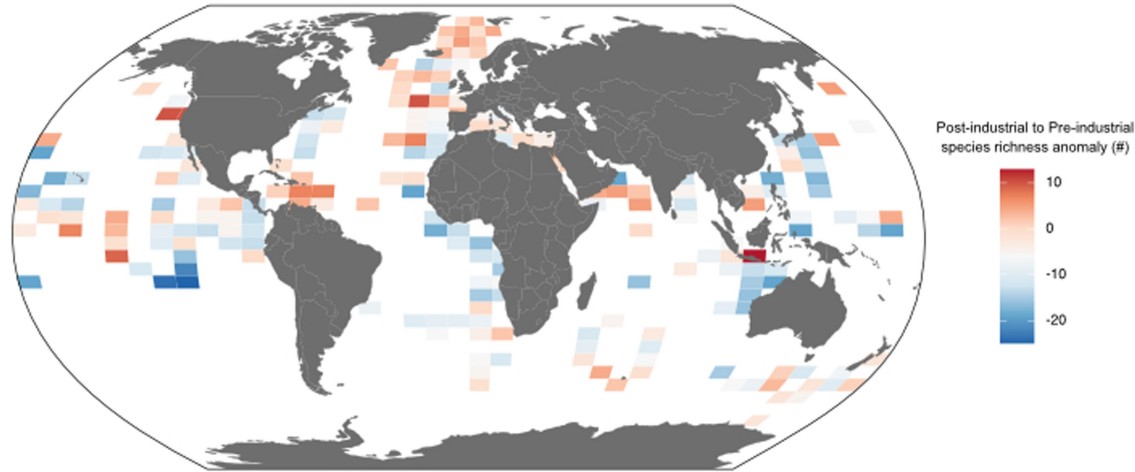

**Extended Data Fig. 3 | Comparison of Diversity of Major Species in FORCIS and ForCenS.** Global map illustrating the gain (positive values, red colours) or loss (negative values, blue colours) in species diversity comparing the FORCIS and ForCenS databases, plotted on a grid map in any 2.8-degrees latitude and 5.6-degrees longitude box.

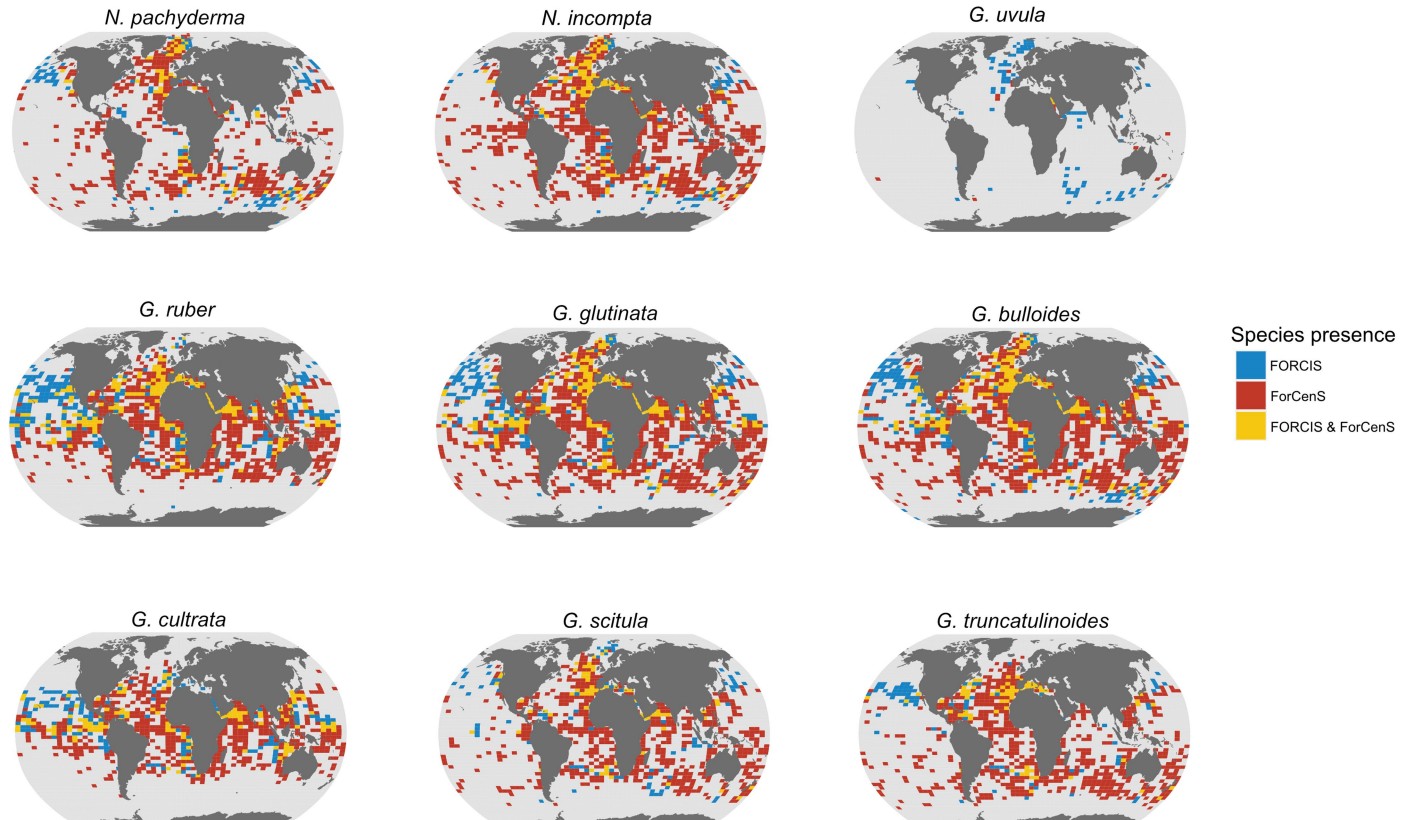

**Extended Data Fig. 4 | Global Distribution of Planktonic Foraminifera Species.** Maps showing the distribution of different PF species in the FORCIS and ForCenS databases. To compare the distribution of the post-industrial (FORCIS) and pre-industrial (ForCenS) samples, the species counts were converted to presence and absence data in both databases using the same taxonomy and plotted on a grid map in any 2.8-degrees latitude and 5.6-degrees longitude box.

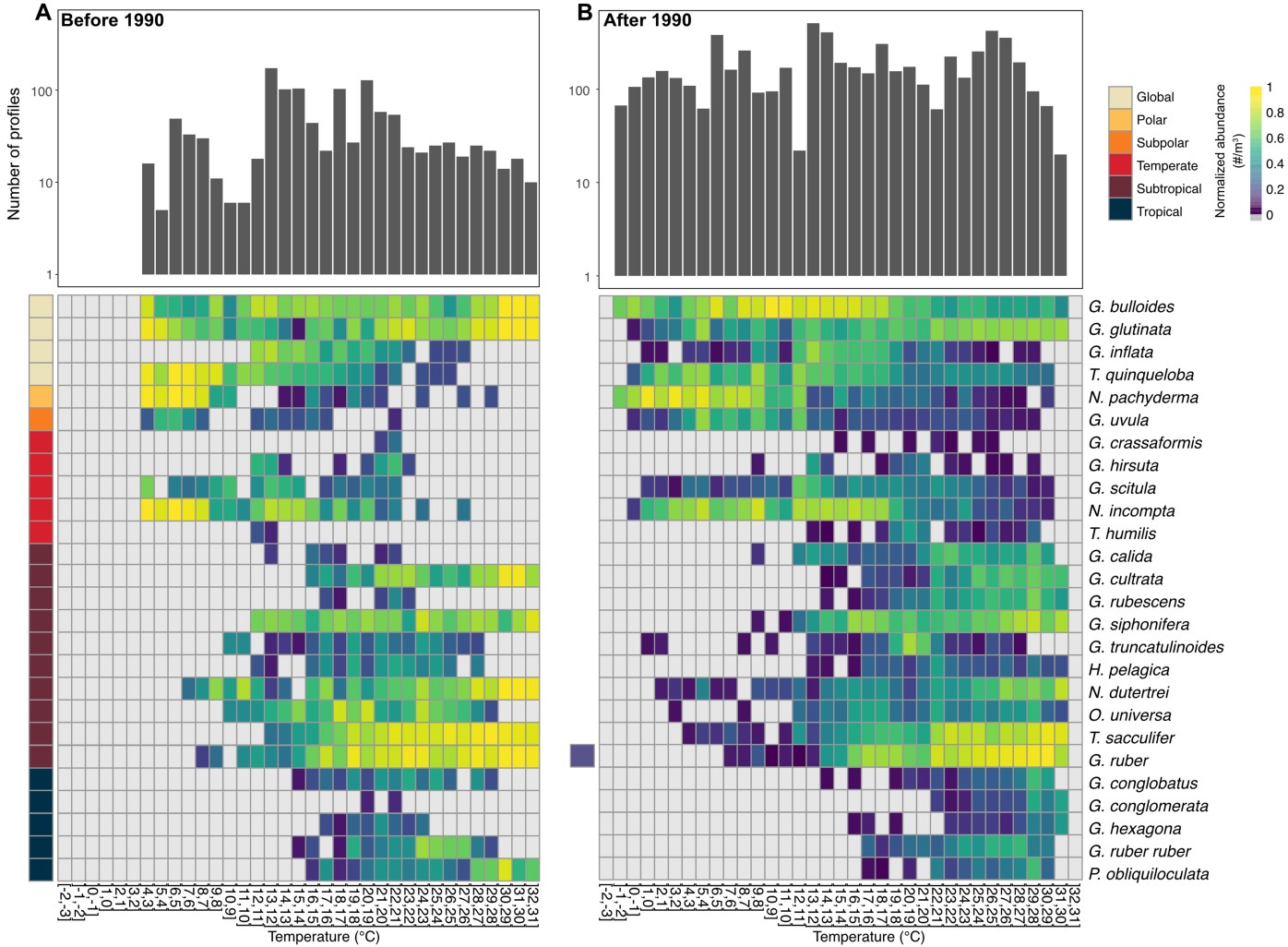

**Extended Data Fig. 5 | Temperature-dependent Variation in Planktonic Foraminifera Abundance.** Normalized abundance variations of the different planktonic Foraminifera species in FORCIS collected using CPR, plankton net, and pump samples from the global ocean across 1 °C-temperature bins in upper 100 m water depth, (**A**) before and (**B**) after 1990. The number of profile_id (contains all the samples collected during the same station, time, and cruise) at each 1 °C-temperature bin is given in the upper panels. To obtain the mean in-situ temperature, the minimum and maximum temperature values were extracted from the Reanalysis Data Hadley EN 4.2.1 analyses g10 dataset, which provides temperature information at a 1°x1° resolution.

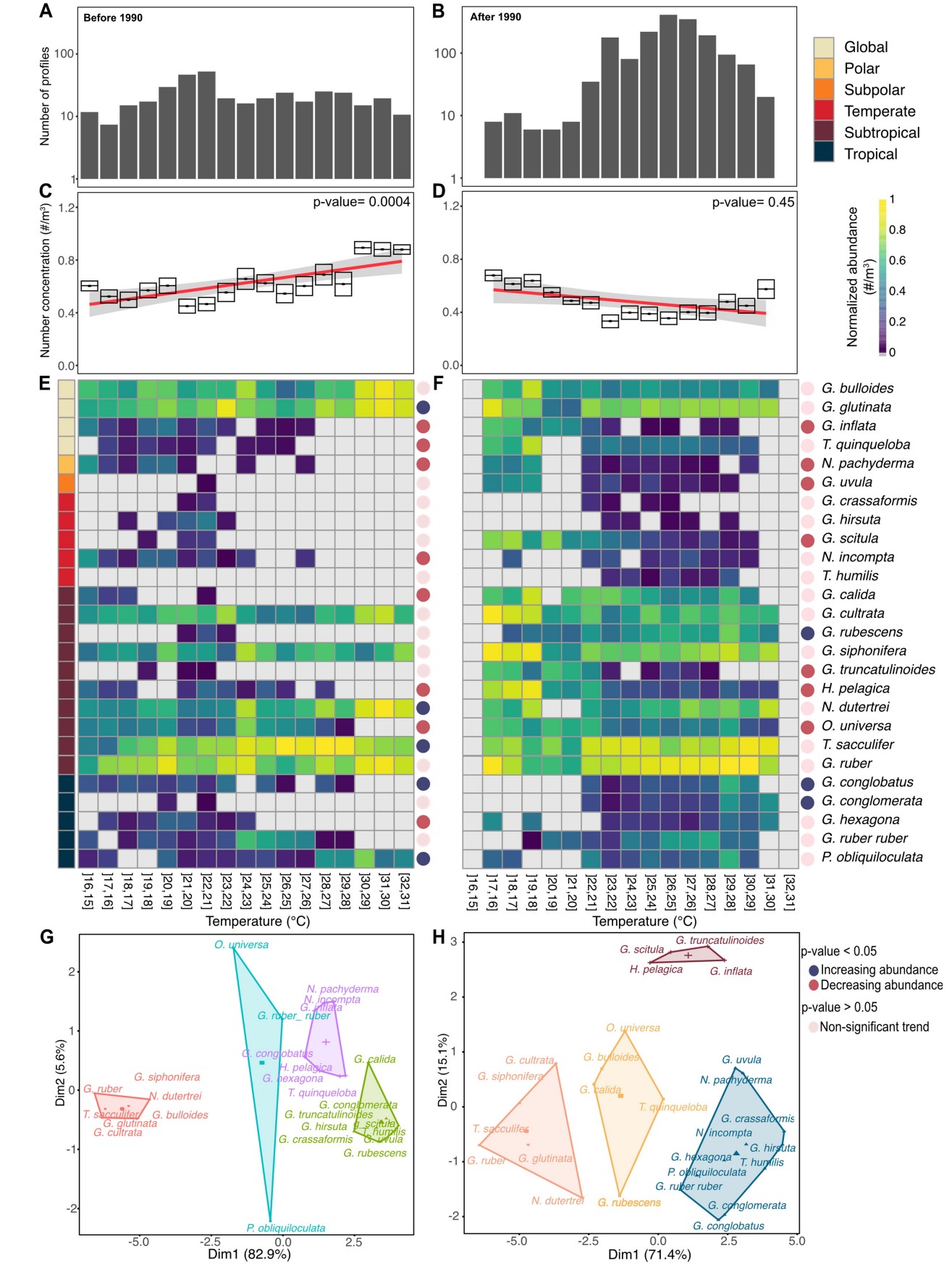

**Extended Data Fig. 6** | See next page for caption.

**Extended Data Fig. 6 | Variations in Abundance in Low Latitude Planktonic Foraminifera Assemblages.** (**A**–**B**) Number of profiles, (**C**–**D**) mean values of the normalized abundance and standard error (represent ±1σ derived and plots are shown as minimum/maximum plots, and the median as a horizontal line) are plotted vs. each 1 °C in-situ temperature bin ranging between 15 °C and 32 °C; calculated for FORCIS samples collected using plankton net during spring and summer, at a depth range from 0 to 100 m, at the low latitudes between 30°S and 30°N, (**A, C**) before and (**B, D**) after 1990, (**E-F**) associated to the normalized abundance variation of each planktonic Foraminifera species vs. binned temperature (each 1 °C) presented in heatmaps for each time period. Statistical significance was assessed using a two-sided one-way ANOVA, with results indicated by large coloured circles: light pink for non-significant trends (P > 0.05), red for statistically significant deepening (P < 0.05), and blue for statistically significant shallowing (P < 0.05). (**G**–**H**) Principal Coordinate Analysis (PCoA) of species assemblage similarities (**G**) before and (**H**) after 1990.

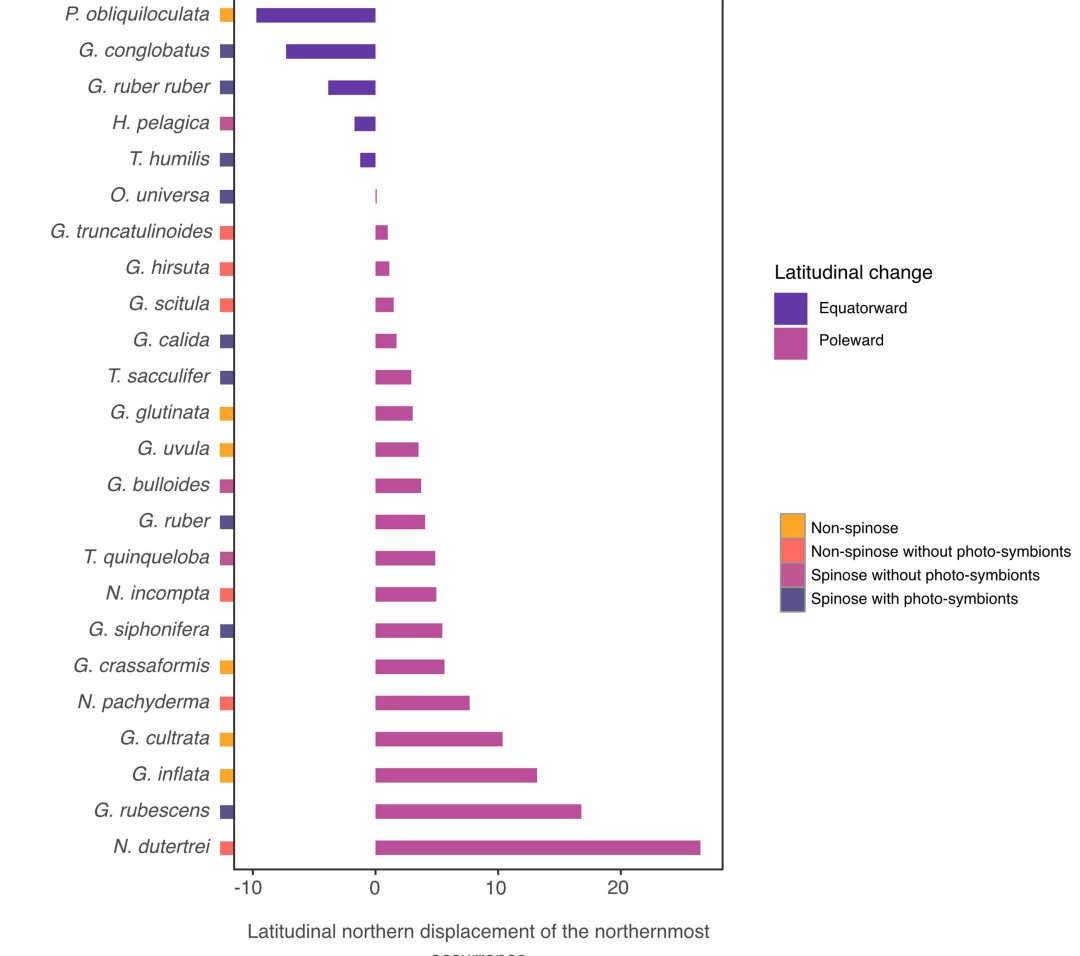

**Extended Data Fig. 7 | Latitudinal Displacement of Planktonic Foraminifera Species.** Latitudinal displacement of planktonic Foraminifera species before and after 1990, according to spring and summer FORCIS data, and over a depth interval from 0 to 100 m in the North Atlantic and Arctic Oceans.

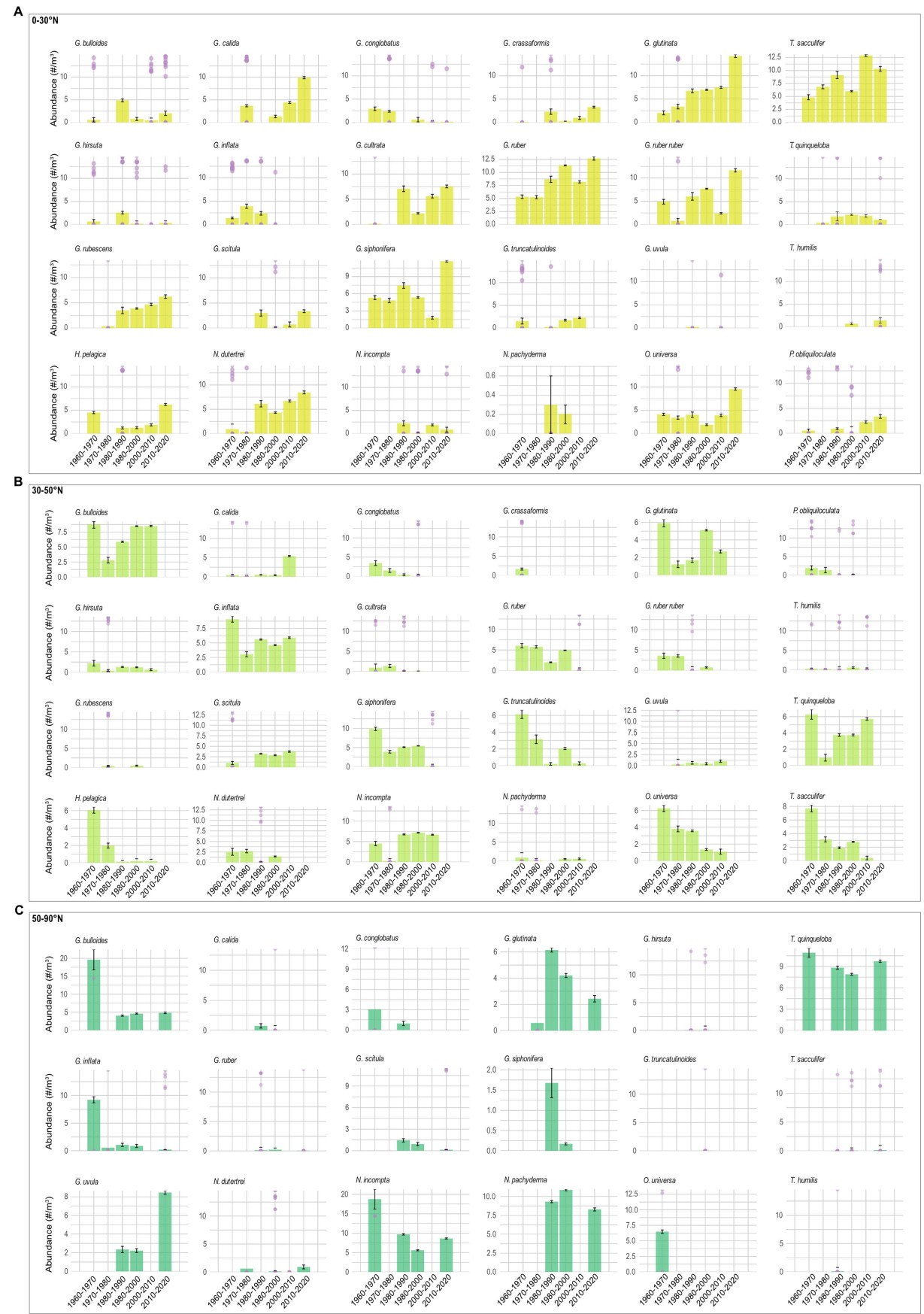

**Extended Data Fig. 8** | See next page for caption.

**Extended Data Fig. 8 | Abundance of Planktonic Foraminifera across Latitudinal Bands.** Abundance of the different planktonic Foraminifera species sourced from FORCIS and collected using the plankton net and pump in the North Atlantic and Arctic Oceans from 0 to 300 m water depth, and over the last decades at different latitudinal bands from (**A**) 0 to 30°N, (**B**) 30 to 50 °N, and (**C**) 50 to 90°N. Error bars represent ±1σ derived and purple dots represent the distribution of the observation when n ≤ 10.

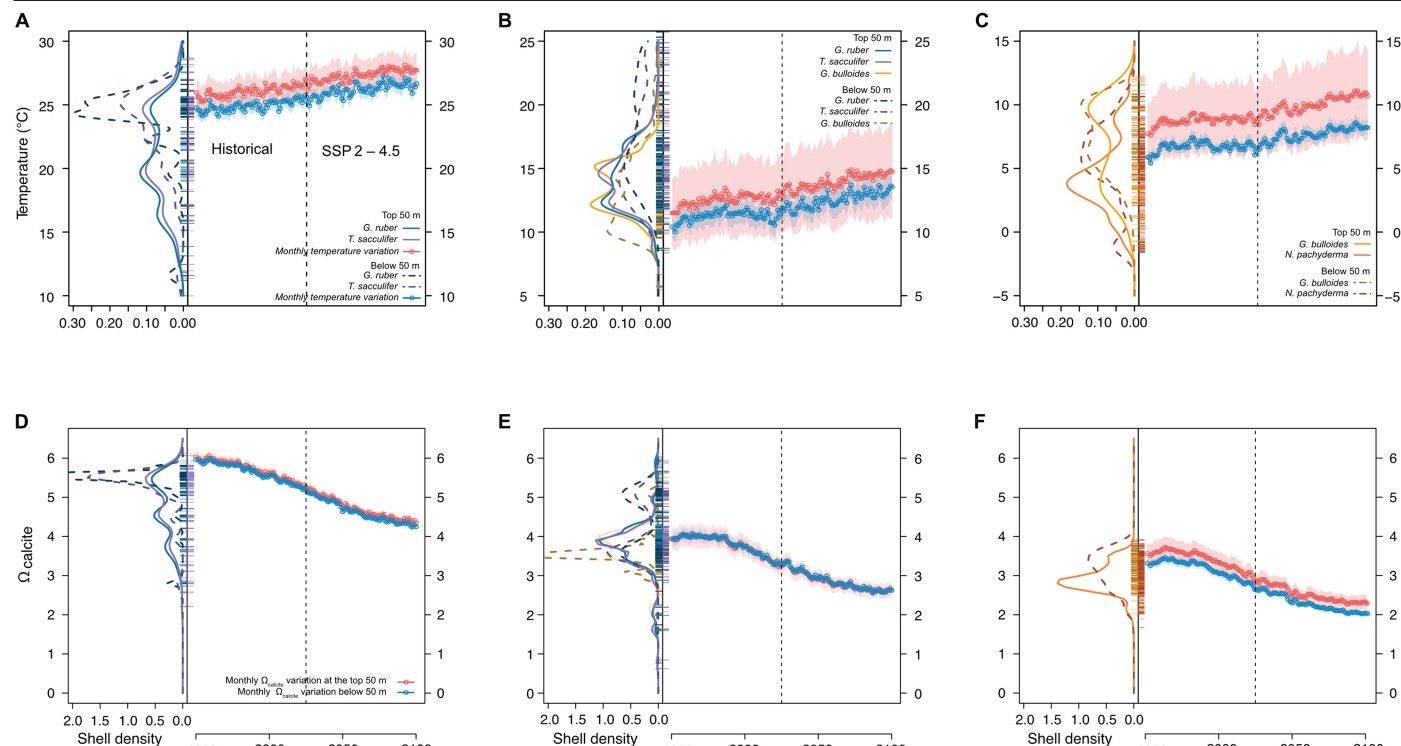

**Extended Data Fig. 9 | Future livability of current Planktonic Foraminifera habitats.** Panels (**A**–**C**): Actual temperature distribution of key planktonic Foraminifera species in the North Atlantic based on the historical data (left panels) and the projected past and future conditions of each zone based on IPSL IPSL-CM6A-LR model output for SSP 2 – 4.5 "middle of the road" climate scenario (right panels). Each model output time series corresponds to an average temperature across the Atlantic basin and in the latitudinal bin with 2 SD shown in the coloured envelope. The marker species are *G. ruber* and *T. sacculifer*, in the tropics (0 to 30°N, panel **A**), *G. ruber*, *T. sacculifer*, and *G. bulloides* in the mid-latitudes (30 to 50°N, panel **B**) and *G. bulloides* and *N. pachyderma* in the high latitudes (50 to 82°N, panel **C**). The vertical distribution and occurrence Foraminifera are shown with solid lines for 0 to 50 m and dashed lines for 50 to 100 m with the cryptic data points plotted on the central axis. Panels **D**–**F** are the same as above but with $\Omega_{calcite}$ on the y-axes.

# Reporting Summary

## Statistics

For all statistical analyses, confirm that the following items are present in the figure legend, table legend, main text, or Methods section.

| n/a | Confirmed | |
|---|---|---|
| ☐ | ☒ | The exact sample size (*n*) for each experimental group/condition, given as a discrete number and unit of measurement |
| ☒ | ☐ | A statement on whether measurements were taken from distinct samples or whether the same sample was measured repeatedly |
| ☐ | ☒ | The statistical test(s) used AND whether they are one- or two-sided *Only common tests should be described solely by name; describe more complex techniques in the Methods section.* |
| ☒ | ☐ | A description of all covariates tested |
| ☐ | ☒ | A description of any assumptions or corrections, such as tests of normality and adjustment for multiple comparisons |
| ☐ | ☒ | A full description of the statistical parameters including central tendency (e.g. means) or other basic estimates (e.g. regression coefficient) AND variation (e.g. standard deviation) or associated estimates of uncertainty (e.g. confidence intervals) |
| ☐ | ☒ | For null hypothesis testing, the test statistic (e.g. *F*, *t*, *r*) with confidence intervals, effect sizes, degrees of freedom and *P* value noted *Give P values as exact values whenever suitable.* |
| ☒ | ☐ | For Bayesian analysis, information on the choice of priors and Markov chain Monte Carlo settings |
| ☒ | ☐ | For hierarchical and complex designs, identification of the appropriate level for tests and full reporting of outcomes |
| ☒ | ☐ | Estimates of effect sizes (e.g. Cohen's *d*, Pearson's *r*), indicating how they were calculated |

*Our web collection on statistics for biologists contains articles on many of the points above.*

## Software and code

Policy information about availability of computer code

| Data collection | The FORCIS database used for this paper is available on Zenodo through https://zenodo.org/record/8186736. ForCenS database is also available from https://doi.pangaea.de/10.1594/PANGAEA.873570. |
|---|---|
| Data analysis | Codes to harmonize the number concentration data were sourced from https://zenodo.org/records/10750545. All codes used for data analysis and generation of figures related to this article can be accessed on Zenodo at https://zenodo.org/records/10881387 The computational analysis conducted in this study utilized a variety of open-source tools in the R environment version 4.1.2 . In the comprehensive analysis of planktonic Foraminifera abundance variations, multiple high-performance R packages were deployed. For handling string manipulations and pattern matching, `stringr` is utilized . `dplyr` allowed for robust data transformation and filtering , while `vegan` conducted ecological multivariate data analyses . Reading and writing Excel files were made seamless using `openxlsx` , whereas phylogenetic and evolutionary analysis was facilitated by `ape` . The package `pheatmap` allows for creating heatmaps . Clustering and partitioning of data to identify patterns were executed using `cluster` , while results from multivariate data analyses were extracted and visualized using `factoextra` . The `viridis` package supplies colorblind-friendly color palettes , and `tidyr` enables easier data cleaning and wrangling . `ggplot2` and `ggpubr` were used for creating high-quality graphics , whereas `reshape2` and `reshape` facilitated the reshaping of data structures . The `plotrix` package provided additional plotting functionalities . Performance and risk calculations were executed with `PerformanceAnalytics` , while correlation matrices were visualized using `ggcorrplot` . For visualization scaling, `scales` was applied . Spatial data visualization was carried out using `ggspatial` and `ggmap` , and geographical maps were drawn using `maps` . The `ggExtra` package enriched `ggplot2` graphics . Lastly, the `gridExtra` package enabled the arrangement of multiple `grid`-based plots . This extensive usage of high-performance R packages significantly contributed to robust, reproducible, and efficient data analysis in this work. |

For manuscripts utilizing custom algorithms or software that are central to the research but not yet described in published literature, software must be made available to editors and reviewers. We strongly encourage code deposition in a community repository (e.g. GitHub). See the Nature Portfolio guidelines for submitting code & software for further information.

## Data

Policy information about [availability of data](availability of data)

All manuscripts must include a [data availability statement](data availability statement). This statement should provide the following information, where applicable:
- Accession codes, unique identifiers, or web links for publicly available datasets
- A description of any restrictions on data availability
- For clinical datasets or third party data, please ensure that the statement adheres to our [policy](policy)

The FORCIS database used for this paper is available on Zenodo through https://zenodo.org/record/8186736.
ForCenS database is also available from https://doi.pangaea.de/10.1594/PANGAEA.873570.

## Research involving human participants, their data, or biological material

Policy information about studies with [human participants or human data](human participants or human data). See also policy information about [sex, gender (identity/presentation), and sexual orientation](sex, gender (identity/presentation), and sexual orientation) and [race, ethnicity and racism](race, ethnicity and racism).

| | |
|---|---|
| Reporting on sex and gender | N/A |
| Reporting on race, ethnicity, or other socially relevant groupings | N/A |
| Population characteristics | N/A |
| Recruitment | N/A |
| Ethics oversight | N/A |

Note that full information on the approval of the study protocol must also be provided in the manuscript.

# Field-specific reporting

Please select the one below that is the best fit for your research. If you are not sure, read the appropriate sections before making your selection.

☐ Life sciences   ☐ Behavioural & social sciences   ☒ Ecological, evolutionary & environmental sciences

For a reference copy of the document with all sections, see [nature.com/documents/nr-reporting-summary-flat.pdf](nature.com/documents/nr-reporting-summary-flat.pdf)

# Ecological, evolutionary & environmental sciences study design

All studies must disclose on these points even when the disclosure is negative.

| | |
|---|---|
| Study description | database analyses of the modern planktonic foraminifera response to environmental changes |
| Research sample | Plankton abundances, habitat (spatial and vertical) and diversity changes |
| Sampling strategy | Analyses of the FORCIS database: holds about 188000 subsamples coming from ~163 000 samples collected from different oceanographic environments by plankton nets (~22 000 subsamples from ~6 000 samples), Continuous Plankton Recorders (CPR) (~157 000 subsamples), sediment trap (~9 000 subsamples), and plankton pump (~400 subsamples) from the global ocean. |
| Data collection | FORCIS database |
| Timing and spatial scale | From 1910 to 2018 |
| Data exclusions | *If no data were excluded from the analyses, state so OR if data were excluded, describe the exclusions and the rationale behind them, indicating whether exclusion criteria were pre-established.* |
| Reproducibility | *Describe the measures taken to verify the reproducibility of experimental findings. For each experiment, note whether any attempts to repeat the experiment failed OR state that all attempts to repeat the experiment were successful.* |
| Randomization | *Describe how samples/organisms/participants were allocated into groups. If allocation was not random, describe how covariates were controlled. If this is not relevant to your study, explain why.* |
| Blinding | *Describe the extent of blinding used during data acquisition and analysis. If blinding was not possible, describe why OR explain why blinding was not relevant to your study.* |

Did the study involve field work?  ☐ Yes  ☐ No

## Field work, collection and transport

Field conditions | *Describe the study conditions for field work, providing relevant parameters (e.g. temperature, rainfall).*

Location | *State the location of the sampling or experiment, providing relevant parameters (e.g. latitude and longitude, elevation, water depth).*

Access & import/export | *Describe the efforts you have made to access habitats and to collect and import/export your samples in a responsible manner and in compliance with local, national and international laws, noting any permits that were obtained (give the name of the issuing authority, the date of issue, and any identifying information).*

Disturbance | *Describe any disturbance caused by the study and how it was minimized.*

# Reporting for specific materials, systems and methods

We require information from authors about some types of materials, experimental systems and methods used in many studies. Here, indicate whether each material, system or method listed is relevant to your study. If you are not sure if a list item applies to your research, read the appropriate section before selecting a response.

### Materials & experimental systems

| n/a | Involved in the study |
|---|---|
| ☐ | ☐ Antibodies |
| ☐ | ☐ Eukaryotic cell lines |
| ☒ | ☐ Palaeontology and archaeology |
| ☐ | ☐ Animals and other organisms |
| ☒ | ☐ Clinical data |
| ☐ | ☐ Dual use research of concern |
| ☐ | ☐ Plants |

### Methods

| n/a | Involved in the study |
|---|---|
| ☒ | ☐ ChIP-seq |
| ☐ | ☐ Flow cytometry |
| ☒ | ☐ MRI-based neuroimaging |

## Antibodies

Antibodies used | *Describe all antibodies used in the study; as applicable, provide supplier name, catalog number, clone name, and lot number.*

Validation | *Describe the validation of each primary antibody for the species and application, noting any validation statements on the manufacturer's website, relevant citations, antibody profiles in online databases, or data provided in the manuscript.*

## Eukaryotic cell lines

Policy information about cell lines and Sex and Gender in Research

Cell line source(s) | *State the source of each cell line used and the sex of all primary cell lines and cells derived from human participants or vertebrate models.*

Authentication | *Describe the authentication procedures for each cell line used OR declare that none of the cell lines used were authenticated.*

Mycoplasma contamination | *Confirm that all cell lines tested negative for mycoplasma contamination OR describe the results of the testing for mycoplasma contamination OR declare that the cell lines were not tested for mycoplasma contamination.*

Commonly misidentified lines (See ICLAC register) | *Name any commonly misidentified cell lines used in the study and provide a rationale for their use.*

## Animals and other research organisms

Policy information about studies involving animals; ARRIVE guidelines recommended for reporting animal research, and Sex and Gender in Research

Laboratory animals | *For laboratory animals, report species, strain and age OR state that the study did not involve laboratory animals.*

Wild animals | *Provide details on animals observed in or captured in the field; report species and age where possible. Describe how animals were caught and transported and what happened to captive animals after the study (if killed, explain why and describe method; if released,*

*(say where and when) OR state that the study did not involve wild animals.*

| | |
|---|---|
| Reporting on sex | *Indicate if findings apply to only one sex; describe whether sex was considered in study design, methods used for assigning sex. Provide data disaggregated for sex where this information has been collected in the source data as appropriate; provide overall numbers in this Reporting Summary. Please state if this information has not been collected. Report sex-based analyses where performed, justify reasons for lack of sex-based analysis.* |
| Field-collected samples | *For laboratory work with field-collected samples, describe all relevant parameters such as housing, maintenance, temperature, photoperiod and end-of-experiment protocol OR state that the study did not involve samples collected from the field.* |
| Ethics oversight | *Identify the organization(s) that approved or provided guidance on the study protocol, OR state that no ethical approval or guidance was required and explain why not.* |

Note that full information on the approval of the study protocol must also be provided in the manuscript.

# Dual use research of concern

Policy information about dual use research of concern

## Hazards

Could the accidental, deliberate or reckless misuse of agents or technologies generated in the work, or the application of information presented in the manuscript, pose a threat to:

No | Yes
- ☐ ☐ Public health
- ☐ ☐ National security
- ☐ ☐ Crops and/or livestock
- ☐ ☐ Ecosystems
- ☐ ☐ Any other significant area

## Experiments of concern

Does the work involve any of these experiments of concern:

No | Yes
- ☐ ☐ Demonstrate how to render a vaccine ineffective
- ☐ ☐ Confer resistance to therapeutically useful antibiotics or antiviral agents
- ☐ ☐ Enhance the virulence of a pathogen or render a nonpathogen virulent
- ☐ ☐ Increase transmissibility of a pathogen
- ☐ ☐ Alter the host range of a pathogen
- ☐ ☐ Enable evasion of diagnostic/detection modalities
- ☐ ☐ Enable the weaponization of a biological agent or toxin
- ☐ ☐ Any other potentially harmful combination of experiments and agents

# Plants

| | |
|---|---|
| Seed stocks | *Report on the source of all seed stocks or other plant material used. If applicable, state the seed stock centre and catalogue number. If plant specimens were collected from the field, describe the collection location, date and sampling procedures.* |
| Novel plant genotypes | *Describe the methods by which all novel plant genotypes were produced. This includes those generated by transgenic approaches, gene editing, chemical/radiation-based mutagenesis and hybridization. For transgenic lines, describe the transformation method, the number of independent lines analyzed and the generation upon which experiments were performed. For gene-edited lines, describe the editor used, the endogenous sequence targeted for editing, the targeting guide RNA sequence (if applicable) and how the editor was applied.* |
| Authentication | *Describe any authentication procedures for each seed stock used or novel genotype generated. Describe any experiments used to assess the effect of a mutation and, where applicable, how potential secondary effects (e.g. second site T-DNA insertions, mosiacism, off-target gene editing) were examined.* |

# Flow Cytometry

## Plots

Confirm that:

☐ The axis labels state the marker and fluorochrome used (e.g. CD4-FITC).

☐ The axis scales are clearly visible. Include numbers along axes only for bottom left plot of group (a 'group' is an analysis of identical markers).

☐ All plots are contour plots with outliers or pseudocolor plots.

☐ A numerical value for number of cells or percentage (with statistics) is provided.

## Methodology

| | |
|---|---|
| Sample preparation | *Describe the sample preparation, detailing the biological source of the cells and any tissue processing steps used.* |
| Instrument | *Identify the instrument used for data collection, specifying make and model number.* |
| Software | *Describe the software used to collect and analyze the flow cytometry data. For custom code that has been deposited into a community repository, provide accession details.* |
| Cell population abundance | *Describe the abundance of the relevant cell populations within post-sort fractions, providing details on the purity of the samples and how it was determined.* |
| Gating strategy | *Describe the gating strategy used for all relevant experiments, specifying the preliminary FSC/SSC gates of the starting cell population, indicating where boundaries between "positive" and "negative" staining cell populations are defined.* |

☐ Tick this box to confirm that a figure exemplifying the gating strategy is provided in the Supplementary Information.

