## [Peer Review File · Nature]

Manuscript Title: Migrating is not enough for modern planktonic Foraminifera in a changing ocean

Reviewer Comments & Author Rebuttals

Reviewer Reports on the Initial Version:

Referees' comments:

Referee #1 (Remarks to the Author):

Modern planktonic Foraminifera: migrating is not enough
By Sonia Chaabane and others

This is an extremely interesting and novel study, bringing together multiple data sets to determine recent and potential future plankton changes. It is widely accepted that anthropogenic warming will have an impact on life on Earth, but analysis and quantification of the effects that have already taken place over the last 50 or so years have been lacking. In this paper, the authors use FORCIS (a database of planktonic foraminifera with data going back to 1910). This manuscript builds on Chaabane et al. (2023) Scientific Data where the FORCIS database was published. The ability to examine how plankton communities have changed vertically, spatially and through time is particularly novel and will be of interest to a wide audience, including marine biologists and paleontologists interested in the macro and micro fauna and flora, and ocean and climate modellers.

Having said that, the text is weakly supported or unclear in places. Presently there are several statements that are unsupported by data or references, and inconsistencies in the text and data tables. Overall, I'm positive about this manuscript and overall analysis of the data which has important implication for the future oceans, but in its current form I was unsatisfied by the quality and robustness of the presentation, and had to keep checking whether I'd missed something or whether the text was inconsistent with itself or the references provided. Below I outline some of the changes needed to progress the manuscript to publication.

For readers interested in biodiversity change, but not so familiar with planktonic foraminifera [PF], it needs to be stated much earlier that PF are strongly temperature controlled, with their maximum abundance at optimum temperatures (e.g., Kucera, 2007). The species-specific temperatures are briefly introduced lines 207-209, but I think this needs to be earlier and with some more detail.

Is the cut-off year 1995 or 1997? Both are listed in the text. 1995 is given as the cut-off date in the methods (line 665) but 1997 in the main text line 391. In Extended Data Figure 6 B it is concentration before and after 1995 plotted. But in the text and figure captions, it is response before and after 1997 that is given (e.g., line 145, 148). In supplementary Table 4 it is 1997 in the caption but 1995 in the

column headings.

Some consideration could be given to the microbial response to temperature, remineralization in the water column and response seen planktonic foraminifera. Boscolo-Galazzo et al. (2021) showed PFs communities occupying shallower depths in warmer intervals due to the remineralization of organic matter in the upper water column. Thus current and future warming would not just have plankton communities shifting polewards (line 70) but also shifting their position within the water column to shallower depths. The link between temperature and food supply is mentioned in lines 123-124, but it is vague simply stating that temperature effects food supply, but is not specific on how. It could be argued that the 'erosion of biodiversity observed in the low latitudes' (line 127) could be due to warming and remineralization of organic matter nearer the surface.

Lines 105-110: This section needs rephrasing. it states species diversity has changed from 4 to 18 species to 3 to 24 species in mid to high latitudes. Some further details and specifics are needed here. It further states (line 109) that modern diversity exceeds pre-industrial by up to 10 species, which is at odds with the lines above. Some details of mean diversity would help here.

Line 113: *Globorotalia scitula* is stated as a mixed-layer dweller but no reference is given. According to mikrotax it is sub-thermocline, Rebotim et al. (2019) and Birch et al. (2013) suggest a deep-dwelling habitat. It records the most positive oxygen isotope values relative to other species in the assemblage, suggesting a deep calcification depth (Birch et al., 2013; Boscolo-Galazzo et al., 2021).

Somewhere (most probably Supplementary Table 1) the full generic name of all species discussed should be given. Currently these are not listed at all. The various genera beginning with G and T are confusing and some species have switched genera in the past as taxonomy has been updated.

In Supplementary Table 1, do the blank spaces in column 3 'Photo-symbionts' mean that the relationship is unknown? It is puzzling as for *G. elongatus* the column is empty, but for *G. ruber albus* or *elongatus* it says Yes, but neither species are mentioned in Takagi et al. (2019) which is the reference for this column. *G. falconensis* also states 'yes' for photo-symbionts, but is also not mentioned in Takagi et al. (2019). Thus the table in its present form is very confusing and inconsistent with the references.

Line 62: A reference is needed for increasing water stratification

Lines 118 and 119: There are two sentences that start 'This' but I lost the thread of what each 'this' refers to.

Line 154: does shallow-water masses mean mixed-layer?

Line 155: I did not understand the reference here to low-turbidity waters, as turbidity is not mentioned elsewhere in the text, does not appear to be measured and is not shown in Fig. 2B (to which the reader is referred to).

Lines 234-250: I found this section too speculative, a summary of plankton community response in other organisms such as nanoplankton, picoplankton, or other macro-organisms would be welcome here.

Line 273 and 389: Change 'this' to 'these'

Lines 330-331: repetition of corresponds, corresponding
Line 380: what is the e after 1990?
Figure captions have inconsistency in US vs UK English spelling.
Line 641: should read Extended data Figure 7.

References

Birch, Heather, et al. "Planktonic foraminifera stable isotopes and water column structure: Disentangling ecological signals." *Marine Micropaleontology* 101 (2013): 127-145.

Boscolo-Galazzo, Flavia, et al. "Temperature controls carbon cycling and biological evolution in the ocean twilight zone." *Science* 371.6534 (2021): 1148-1152.

Kucera, M., 2007. Planktonic foraminifera as tracers of past oceanic environments. *Developments in marine geology* 1, 213-262.

Rebotim, A., Voelker, A. H. L., Jonkers, L., Waniek, J. J., Schulz, M., and Kucera, M.: Calcification depth of deep-dwelling planktonic foraminifera from the eastern North Atlantic constrained by stable oxygen isotope ratios of shells from stratified plankton tows, *J. Micropalaeontol.*, 38, 113–131, <https://doi.org/10.5194/jm-38-113-2019>, 2019.

Takagi, H. et al. Characterizing photosymbiosis in modern planktonic foraminifera. *546 Biogeosciences* 16, 3377–3396 (2019).

Referee #2 (Remarks to the Author):

In this paper, the authors compare foraminifera samples/assemblage data from continuous plankton recorders, plankton nets, pumps, and sediment traps (ForCIS) with foraminifera assemblage data from ForCenS (seafloor sediment assemblages). The paper is well written.

Generally, my summary is that the paper is trying to do too much, and I do not think the findings are suitable for publication in the current format. The ForCenS data and FORCIS datasets do not have the same coverage and the data included in the figures is inconsistent in both time and space. I didn't understand why in some figures date cutoffs were chosen vs. a different date cutoff in another figure. There are too many figures for a short format paper with not enough detail provided for me to fully understand what data was used and how the figures and conclusions were drawn. Their key finding: that migration is not enough, is probably accurate, as noted in other previously published papers (Roy et al., 2015; Jonkers et al., 2019).

I'm not sure I can provide detailed suggestions for what the authors can do to streamline the text or their findings, but I have included some concerns below and comments for the figures.

A few major concerns:

1. Is it appropriate to compare, for example: plankton pump and CPR data that provide 'snapshots' of assemblages with seafloor sediments samples that are time integrated. I would argue that comparing sediment trap data with seafloor sediment samples is appropriate, but that has already been done by Jonkers et al.,. For example: Figure 1 suggests that the species richness has decreased and is much lower in the modern net, pump, and cpr data compared to ForCenS. But isn't that to be expected because the data are completely different? For one, CPR and pumps only obtain samples from the top 10 meters of the water column and most foraminifera are found much deeper in the water column, and so naturally the species richness will be lower in the pump and CPR samples. Note that the figure panels are missing panel letters (A) and (B).
2. The cutoff dates chosen throughout the paper are justified by the authors somewhat, but also seem arbitrary. 1997 is a cutoff used in figure 2, but in extended figure 5 the data are binned by <1993, 1993-2000, and >2000. In another paragraph, the cutoff date was 1995.
3. Why are some species discussed in detail? For example, in the paragraph beginning on line 111 the authors discuss a poleward shift of several species. The extended data figure shows 9 species. Are these the only species that show the poleward shift? Why are only 9 species included in Extended figure 4. The text states that *G. scitula* is notable in that it has expanded its modern ecological niche, now appearing at high latitudes above 80N. Couldn't this finding simply be due to sampling bias? Why aren't the other species discussed in the manuscript since they are included in the figure?
4. Extended figure 5, discussed on line 115, is confusing. I don't know what the histogram axis means: number of profile_id. The y-axis is, I think, temperature, but there is no label. Why are the data binned by <1993, 1993-2000, and >2000? The previous paragraph used 1995 as a cutoff. Are the data obtained from latitudinal bins?

Figure comments:

Figure 1:

Missing panel labels. Pump and CPR data cannot be compared to sediment samples (ForCenS) because the pump/CPR data samples the surface ocean whereas the latter is a time integration of samples, sometimes significantly long, depending on the sedimentation rate. It might be appropriate to compare the sediment trap samples with the seafloor samples.

Figure 2:

I don't understand this figure. Is the width of the colored bands related to abundances? Why is before and after 1997 chosen for the cutoff? Is the transition to deeper water for the NS species driven by temperature or perhaps food availability? For example, crassiformis deepening by 45 meters because it was found in one location before and after 1997 at different depths? Were the conditions the same? Or did it migrate because hydrographic conditions changed its preferred thermal tolerance?

Figure 3: I have some concerns that there is significant sampling bias in these results. For example, the data from the 1940-1950 is from one location off the coast of N. America. With each decade, the number of locations sampled increased significantly, and thus do the normalized number concentration

per cubic meter go down simply because the sampled area went up?

Figure 4: There is not enough detail in the paper, the figures, or the figure captions for me to fully grasp the findings presented in this figure.

Minor note: The Fox paper erroneously used a thinly calcified (likely live at the time of collection) *N. dutertrei* and thus the 74% thinner modern specimen is simply due to comparing two different ontogenic stages (a live vs. dead foram), not OA.

Referee #3 (Remarks to the Author):

The manuscript titled 'Modern planktonic Foraminifera: migrating is not enough' by Chaabane et al., is an exceptional and highly topical piece of work. The manuscript is elegantly written with a broader audience in mind and therefore will be a cornerstone for future research and public discussions.

For the first time, horizontal and vertical distribution patterns were plotted using the invaluable dataset from ForCenS global census. The authors used this long-term dataset in a novel way to understand how planktonic foraminifer have migrated through space and time. The authors also took it one step future and projected possible migration patterns considering the two most prominent physico-chemical parameters of climate change influencing marine calcifiers (calcite saturation and temperature). This work provides an unprecedented look at the nuanced adaptive strategies of a vital ecosystem engineer that will have a significant impact on climate regulation into the future.

The methods and interpretations of the data are appropriate and robust. The figures are synthesised and presented in an engaging way with ample explanation in the extended data section. The caveats, particularly the number of observations, are mentioned and dealt with honestly. There are only a few typos and minor edits that need to be corrected (e.g., Sup Table 4 G. hexagona p values should have an *).

The quality of this global study is outstanding and its' original conclusions will have a far reaching influence beyond the marine micropaleontological community, therefore, I highly recommend this work be published in Nature.

Author Rebuttals to Initial Comments:

27-03-2024

COMMENTS TO THE AUTHORS OF NATURE MANUSCRIPT 2023-10-19099 (“MODERN PLANKTONIC FORAMINIFERA: MIGRATING IS NOT ENOUGH”) WITH POINT-BY-POINT RESPONSES FROM THE AUTHORS (IN BLUE)

Your manuscript, "Modern planktonic Foraminifera: migrating is not enough", has now been seen by 3 referees, whose comments are attached below. While they find your work of potential interest, as do we, they have raised important concerns that in our view need to be addressed before we can consider publication in Nature.

Should further experiments allow you to fully address these criticisms, we would be happy to consider a revised manuscript (unless something similar has been accepted at Nature or appeared elsewhere in the meantime). In addition, I would like to emphasize that we agree with Referee #2 that the manuscript should be revised towards a better organization/presentation of the data. If needed and justified, we can give the paper more Extended Data figures, should it make it to publication. In the revised version, we may also solicit the input of a fourth reviewer. We hope to receive your revised paper within four to six months. If you cannot complete the required revisions within this time frame, please let us know when you would anticipate being able to submit a revised manuscript.

Authors: We would like to thank the editors and the three anonymous reviewers for taking the time to review our manuscript and for their supportive comments. In accordance with the comments from the reviewers, we made changes throughout the text, analysis, figures, tables and the supplementary material.

Please find below our point-by-point responses to the anonymous reviewers' comments. We made sure to carefully address every comment in the responses and in the text and figures when needed. We hope that the present revised version will be considered suitable for publication in Nature.

Referee #1

Modern planktonic Foraminifera: migrating is not enough
By Sonia Chaabane and others

This is an extremely interesting and novel study, bringing together multiple data sets to determine recent and potential future plankton changes. It is widely accepted that anthropogenic warming will have an impact on life on Earth, but analysis and quantification of the effects that have already taken place over the last 50 or so years have been lacking. In this paper, the authors use FORCIS (a database of planktonic foraminifera with data going back to 1910). This manuscript builds on Chaabane et al. (2023) Scientific Data where the FORCIS database was published. The ability to examine how plankton communities have changed vertically, spatially and through time is particularly novel and will be of interest to a wide audience, including

marine biologists and paleontologists interested in the macro and micro fauna and flora, and ocean and climate modellers.

Having said that, the text is weakly supported or unclear in places. Presently there are several statements that are unsupported by data or references, and inconsistencies in the text and data tables. Overall, I'm positive about this manuscript and overall analysis of the data which has important implication for the future oceans, but in its current form I was unsatisfied by the quality and robustness of the presentation, and had to keep checking whether I'd missed something or whether the text was inconsistent with itself or the references provided. Below I outline some of the changes needed to progress the manuscript to publication.

Authors: We would like to thank the reviewer for their constructive and helpful review. We have worked on the manuscript to make it clearer. We have also made sure that all statements are now clearly and solidly backed up by relevant references, analyses and figures.

For readers interested in biodiversity change, but not so familiar with planktonic foraminifera [PF], it needs to be stated much earlier that PF are strongly temperature controlled, with their maximum abundance at optimum temperatures (e.g., Kucera, 2007). The species-specific temperatures are briefly introduced lines 207-209, but I think this needs to be earlier and with some more detail.

Reply: We agree with the reviewer that PF are strongly temperature controlled (directly and indirectly), since this impacts their ecology and food availability, thus we added these explanations to the revised version of the text:

“The presence and abundance patterns of PF are significantly influenced by temperature¹⁸.”, in lines 77-78.

“The spatial distribution of PF is strongly controlled by temperature, as their maximum abundance generally matches some temperature optimum^{18,28,42}. The observed sensitivity of PF to temperature contributes to the establishment of a pronounced latitudinal diversity gradient (LDG)^{19,31,43}, and causes the discernible response to ongoing global warming reported here, as for other plankton groups^{30,44}.”, in lines 213-217.

Is the cut-off year 1995 or 1997? Both are listed in the text. 1995 is given as the cut-off date in the methods (line 665) but 1997 in the main text line 391. In Extended Data Figure 6 B it is concentration before and after 1995 plotted. But in the text and figure captions, it is response before and after 1997 that is given (e.g., line 145, 148). In supplementary Table 4 it is 1997 in the caption but 1995 in the column headings.

Reply: We thank the reviewer for pointing this out, and we apologize for the confusion. We have modified all cut-off years to 1990, updated the text and figures (Extended Figs. 5 and 6) accordingly; this did not affect our results. The only exception is in Fig. 2, which investigates the vertical dimension, where we used only multinet data that were first deployed in 1989 in the Atlantic Ocean. In this specific case, to have a balanced datasets before and after, a cut-off year set at 1997, as shown in the figure R1 below. Thus, for the multinet-based data analysis that is plotted in Fig. 2, we use 1997 as the cut-off year.

Fig. R1: Number of samples per year in the low (from 0 to 30°N) and mid (30 to 50°N) latitudes, collected with plankton nets during spring and summer from the North Atlantic and Arctic Oceans, between 0 to 200 m water depth.

Extended Figs. 5 and 6, the main text and the methods have been updated to reflect the change to an homogeneous cut-off year of 1990 (except for Fig. 2). This has not affected the results. In the methods, we now write (lines 351-354): **“To separate the dataset in time, into two fractions of equal size, a cut-off date was set to 1990. In figure 2, however, a different cut-off year is used, set at 1997. That is because this figure is based solely on multinet data, which have only been available since 1989. This allows the comparison of two datasets with similar amounts of information.”**

Some consideration could be given to the microbial response to temperature, remineralization in the water column and response seen planktonic foraminifera. Boscolo-Galazzo et al. (2021) showed PFs communities occupying shallower depths in warmer intervals due to the remineralization of organic matter in the upper water column. Thus current and future warming would not just have plankton communities shifting polewards (line 70) but also shifting their position within the water column to shallower depths. The link between temperature and food supply is mentioned in lines 123-124, but it is vague simply stating that temperature effects food supply, but is not specific on how. It could be argued that the ‘erosion of biodiversity observed in the low latitudes’ (line 127) could be due to warming and remineralization of organic matter nearer the surface.

Reply: We thank the reviewer for raising this important point that is also supported by the new paper of Wang et al. (2023; Nature). Their findings highlight the pivotal role of remineralization of organic matter in the upper water column (100 m depth). Wang et al.'s study emphasizes the intricate link between temperature, organic matter remineralization, and the dynamics of planktonic communities, and we have incorporated these insights into our revised manuscript. Unfortunately, data on the food availability are not available in the same detail as the FORCIS data on planktonic Foraminifera. We cannot assume that temperature only drives PF diversity but it can partly explain the depth habitat of species in the mid-latitude. We clarified our reasoning and complemented it, in the direction suggested by the reviewer, with these two sentences in the revised version:

“These effects are exacerbated for organisms producing a calcium carbonate shell or skeleton, as acidification impedes calcification faster than warming favours it⁷, due to processes such as remineralization of the organic matter in the upper water column^{8,9}.”, in the introduction in lines 63-66.

“The lack of an evident deepening depth habitat of some PF in low to mid latitudes might be linked to remineralization processes nearer the ocean surface, as suggested by recent findings⁸. Elevated temperatures stimulate microorganism metabolism, expediting the degradation of organic matter. This process results in the availability of a more accessible food source near the ocean surface, potentially influencing the distribution patterns of specific PF species^{8,9}.”, in lines 170-175.

Lines 105-110: This section needs rephrasing. It states species diversity has changed from 4 to 18 species to 3 to 24 species in mid to high latitudes. Some further details and specifics are needed here. It further states (line 109) that modern diversity exceeds pre-industrial by up to 10 species, which is at odds with the lines above. Some details of mean diversity would help here.

Reply: We agree with the reviewer. While the initial comparison hinges on diversity extracted from each profile in a wide spatial distribution, the representation in Extended Data Fig. 3 relies on aggregated FORCIS data profiles within a spatial box encompassing 2.8 degrees of latitude and 5.6 degrees of longitude. Consequently, we have opted to concentrate on a more specific region in Figure 1B, where the accumulation of diversity is more pronounced to make the comparison possible with the Extended Data Fig. 3.

“Over the last century, species diversity in each data profile in the mid to high latitudes slightly exceeds pre-industrial levels by four species, increasing from 9 species in pre-industrial data to 13 in post-industrial data (between 65° to 80°N, Fig. 1B). In some regions in the North Atlantic and Pacific Ocean, modern diversity surpasses pre-industrial levels by up to ten species (Extended Data Fig. 3).” in lines 108-112.

Line 113: *Globorotalia scitula* is stated as a mixed-layer dweller but no reference is given. According to mikrotax it is sub-thermocline, Rebotim et al. (2019) and Birch et al. (2013) suggest a deep-dwelling habitat. It records the most positive oxygen isotope values relative to other species in the assemblage, suggesting a deep calcification depth (Birch et al., 2013; Boscolo-Galazzo et al., 2021).

Reply: We agree that *G. scitula* is dwelling in upper sub-thermocline waters, which has first been published by Ottens (1992). We corrected this statement in the revised version and we write now:

“Notably, *G. scitula*, a sub-thermocline dweller^{24–26} in the temperate oceans has expanded its ecological niche in FORCIS, now appearing at high latitudes (80°N) (Extended Data Figs. 4 and 5).”, in lines 118-120.

Somewhere (most probably Supplementary Table 1) the full generic name of all species discussed should be given. Currently these are not listed at all. The various genera beginning with G and T are confusing and some species have switched genera in the past as taxonomy has been updated.

Reply: The full generic name of all species is given now in the revised version of Supplementary Table 1, using the taxonomy revised by Brummer and Kucera 2022.

In Supplementary Table 1, do the blank spaces in column 3 ‘Photo-symbionts’ mean that the relationship is unknown? It is puzzling as for *G. elongatus* the column is empty, but for *G. ruber albus* or *elongatus* it says Yes, but neither species are mentioned in Takagi et al. (2019) which is the reference for this column. *G. falconensis* also states ‘yes’ for photo-symbionts, but is also not mentioned in Takagi et al. (2019). Thus the table in its present form is very confusing and inconsistent with the references.

Reply: We corrected the table S1 for *G. elongatus* and *G. ruber albus* or *elongatus* are both symbionts bearing taxa (Morard et al. 2019). Whereas Takagi did not analyse *G. falconensis*, possibly because she did not study it, *G. falconensis* is well known to host photosymbionts (e.g., Hemleben et al. 1989). We have now added the information to the revised manuscript in the **table S1**.

Line 62: A reference is needed for increasing water stratification

Reply: We added the following reference to the revised version, line **63**:

Scott C. Doney, Mary Ruckelshaus, J. Emmett Duffy, James P. Barry, Francis Chan, Chad A. English, Heather M. Galindo, Jacqueline M. Grebmeier, Anne B. Hollowed, Nancy Knowlton, Jeffrey Polovina, Nancy N. Rabalais, William J. Sydeman, and Lynne D. Talley (2012). "Climate Change Impacts on Marine Ecosystems." Annual Review of Marine Science, 4, 11-37. DOI: 10.1146/annurev-marine-041911-111611

Lines 118 and 119: There are two sentences that start ‘This’ but I lost the thread of what each ‘this’ refers to.

Reply: Corrected in the revised version of the manuscript as following (lines **123-126**):

“The post-industrial PF diversity decrease mirrors a decline in equatorial diversity in the pre-industrial era²⁷, and diminished diversity in warmer low-latitude waters affecting PF distribution²⁸. The observed latitudinal diversity shift in FORCIS is consistent with long-term changes in plankton assemblages, as predicted by several modeling studies^{20,29,30}.”

Line 154: does shallow-water masses mean mixed-layer?

Reply: Here we meant to use a qualitative term to mean surface and mixed layer depths, but realise that it is ambiguous. Therefore, we have rephrased the sentence (now lines **162-165**):

“In the mid-latitudes, tropical and subtropical symbiont bearing species thrive in near surface environments in the mixed layer (0-70m), following the latitudinal shift of surface conditions towards oligotrophic waters; this leads to a majority of these species shoaling over our time series.”

Line 155: I did not understand the reference here to low-turbidity waters, as turbidity is not mentioned elsewhere in the text, does not appear to be measured and is not shown in Fig. 2B (to which the reader is referred to).

Reply: This was indeed a “shortcut”. Here we used the turbidity as a proxy of productivity: we meant oligotrophic waters for clear waters. We deleted “Fig. 2B” and we changed “clear waters” by **“oligotrophic waters”**, in line 164.

Lines 234-250: I found this section too speculative, a summary of plankton community response in other organisms such as nanoplankton, picoplankton, or other macro-organisms would be welcome here.

Reply: This section is indeed partly speculative: we discuss how ecological niches could evolve based on our knowledge of PF ecology, actual PF distribution data, and temperature and water chemistry model projections. To broaden the discussion to other organisms across the food chain, we added the following:

“The consequences of acidification may affect organisms at various levels across the food chain. Photosynthetic calcareous nanoplankton (e.g., coccolithophores) show marked patterns of decreased calcification upon acidification⁵⁶. For other groups, such as picoplankton (e.g. cyanobacteria, small eukaryotes), echinoderms, crustaceans or cephalopods, the calcification response is more nuanced and mediated by phenotypic plasticity^{57,58}. Nevertheless, for most calcifying groups as well as certain fish groups that produce highly soluble forms of calcium carbonate⁵⁹, acidification effects may be particularly detrimental at early-life stages^{57,60}. Overall, calcifying community response will be even more complex, due to conflicting influences of changing nutrients, temperature, and other environmental parameters.”, in lines 278-286.

Line 273 and 389: Change ‘this’ to ‘these’

Reply: Corrected, line 310.

Lines 330-331: repetition of corresponds, corresponding

Reply: “corresponding” changed to **“of”** line 369.

Line 380: what is the e after 1990?

Reply: Corrected, line 427.

Figure captions have inconsistency in US vs UK English spelling.

Reply: Figure caption 2 corrected, lines 646-654.

Line 641: should read Extended data Figure 7.

Reply: “**Extended Data**” is added to the caption, line 727.

References

Birch, Heather, et al. "Planktonic foraminifera stable isotopes and water column structure: Disentangling ecological signals." *Marine Micropaleontology* 101 (2013): 127-145.

Boscolo-Galazzo, Flavia, et al. "Temperature controls carbon cycling and biological evolution in the ocean twilight zone." *Science* 371.6534 (2021): 1148-1152.

Kucera, M., 2007. Planktonic foraminifera as tracers of past oceanic environments. *Developments in marine geology* 1, 213-262.

Rebotim, A., Voelker, A. H. L., Jonkers, L., Waniek, J. J., Schulz, M., and Kucera, M.: Calcification depth of deep-dwelling planktonic foraminifera from the eastern North Atlantic constrained by stable oxygen isotope ratios of shells from stratified plankton tows, *J. Micropalaeontol.*, 38, 113–131, <https://doi.org/10.5194/jm-38-113-2019>, 2019.

Takagi, H. et al. Characterizing photosymbiosis in modern planktonic foraminifera. *546 Biogeosciences* 16, 3377–3396 (2019).

Referee #2

In this paper, the authors compare foraminifera samples/assemblage data from continuous plankton recorders, plankton nets, pumps, and sediment traps (ForCIS) with foraminifera assemblage data from ForCenS (seafloor sediment assemblages). The paper is well written.

Generally, my summary is that the paper is trying to do too much, and I do not think the findings are suitable for publication in the current format. The ForCenS data and FORCIS datasets do not have the same coverage and the data included in the figures is inconsistent in both time and space. I didn't understand why in some figures date cutoffs were chosen vs. a different date cutoff in another figure. There are too many figures for a short format paper with not enough detail provided for me to fully understand what data was used and how the figures and conclusions were drawn. Their key finding: that migration is not enough, is probably accurate, as noted in other previously published papers (Roy et al., 2015; Jonkers et al., 2019).

Authors: We would like to thank the reviewer #2 for the helpful comments and suggestions.

I'm not sure I can provide detailed suggestions for what the authors can do to streamline the text or their findings, but I have included some concerns below and comments for the figures.

A few major concerns:

1. Is it appropriate to compare, for example: plankton pump and CPR data that provide 'snapshots' of assemblages with seafloor sediments samples that are time integrated. I would argue that comparing sediment trap data with seafloor sediment samples is appropriate, but that has already been done by Jonkers et al.,. For example: Figure 1 suggests that the species richness has decreased and is much lower in the modern net, pump, and cpr data compared to ForCenS. But isn't that to be expected because the data are completely different? For one, CPR and pumps only obtain samples from the top 10 meters of the water column and most foraminifera are found much deeper in the water column, and so naturally the species richness will be lower in the pump and CPR samples. Note that the figure panels are missing panel letters (A) and (B).

Reply: We thank the reviewer for the thoughtful comments, and apologize for the missing panels letters (fixed in this revised version). In our study, we acknowledge the inherent obvious differences between plankton pump and CPR data, which provide 'snapshots' of assemblages, and seafloor sediment samples that represent time-integrated data, yet are also affected by multiple preservation/advection issues. For the issue raised by the reviewer, we focus on the presence and absence of species rather than quantitative fluxes and we aim to explore the shape of species diversity patterns rather than temporal assemblage change as in Jonkers et al (2019) study.

The main thrust of this manuscript is therefore to go beyond the spatially discrete analysis made on the sediment traps as in Jonkers et al, to a more integrative study including all threads of evidence, including plankton nets data, which sample Foraminifera over their full living depths. We also note for example that the fact that CPR would underestimate the diversity compared to plankton net sampling has been shown incorrect in the Meilland et 2016 study. The diversity reported in Fig. 1 reflects the presence or absence of individuals regardless of the sampling technique, hence it should account for the vast majority of planktonic Foraminifera.

Nevertheless, in this revision we have added a robust spatial analysis of the database that leverages all sample devices (including sediment traps), and therefore compensate for the inherent temporal sampling biases of nets, pumps and CPRs. We spatially gridded the database

into 10° latitudinal bands and used the 95th percentile of species richness for all devices. Then fitted with a Generalized Additive Model (GAM) to capture the variability in species richness along the latitudinal gradient. With this approach we believe that these metrics will estimate the minimal diversity in each latitudinal band (Fig. 1B).

“Then, in each 10° latitude bin, the 95th percentile of species richness was selected, assuming discrete sampling using nets, CPRs and pumps. These selected data points were then fitted with a Generalized Additive Model (GAM) to capture the variability in species richness along the latitudinal gradient.” in lines 376-379.

We are also cautious about the conclusion as written in this paragraph:

“Though it must be noted that both diversity estimates are associated with a number of biases (e.g., preservation, ontogeny, seasonality etc.), qualitatively, the comparison shows a modern diversity that is higher in high latitudes and lower near the Equator, relative to the preindustrial.” in lines 112-115.

To further clarify the different sampling techniques used in our dataset, we added histograms in Fig. 1 that describe the number of total observations in FORCIS and ForCenS together for each 5-degree latitudinal bin and each species richness level.

We have added panel letters in Fig. 1, as suggested by the reviewer.

Fig. 1

2. The cutoff dates chosen throughout the paper are justified by the authors somewhat, but also seem arbitrary. 1997 is a cutoff used in figure 2, but in extended figure 5 the data are binned by <1993, 1993-2000, and >2000. In another paragraph, the cutoff date was 1995.

Reply: We agree with the reviewer that the choice of cut-off years was not well explained or justified; this was also noted by the first reviewer (see comment). We have modified all cut-off years to 1990, updated the text and figures accordingly; this did not affect our results. The cut-off year for the vertical trend distribution in Fig. 2 is still 1997, because it plots multinet-based data only, and multinet data have only been available since 1989, unlike the rest of the dataset that covers much broader periods.

3. Why are some species discussed in detail? For example, in the paragraph beginning on line 111 the authors discuss a poleward shift of several species. The extended data figure shows 9 species. Are these the only species that show the poleward shift? Why are only 9 species included in Extended figure 4. The text states that *G. scitula* is notable in that it has expanded its modern ecological niche, now appearing at high latitudes above 80N. Couldn't this finding simply be due to sampling bias? Why aren't the other species discussed in the manuscript since they are included in the figure?

Reply:

We agree with the reviewer that the choice of selecting 9 out of the 26 species discussed for this figure was not clearly justified. In the revised version, we explain why we decided to show an example of only these species as they are main environmental niches, extremely used in paleoceanography and show clearly new results (*G. uvula*). Their post-industrial distribution compared to preindustrial distribution is of interest to a wider community as they are now beyond their preindustrial range.

“Then, to contrast the distribution patterns between post-industrial (FORCIS) and pre-industrial (ForCenS) samples, species counts were transformed into presence and absence data, ensuring uniform taxonomy across both databases. These data were then visualized on a grid map, with each grid cell representing 2.8 degrees of latitude and 5.6 degrees of longitude. From the 26 species under consideration, nine were specifically chosen based on their main environmental niches (including polar and subpolar, tropical to subtropical, globally distributed, and deep-sea species), significant relevance to paleoceanography, and the introduction of novel insights (e.g., *G. uvula*; Extended Data Fig. 4).” in lines 389-396.

In the Extended figure 4, *G. scitula* is seen in many samples in FORCIS when missing in ForCenS, while the database reaches these latitudes.

4. Extended figure 5, discussed on line 115, is confusing. I don't know what the histogram axis mean: number of profile_id. The y-axes are, I think, temperature, but there is no label. Why are the data binned by <1993, 1993-2000, and >2000? The previous paragraph used 1995 as a cutoff. Are the data obtained from latitudinal bins?

Reply: We thank the reviewer for raising this point. The 4. Extended data figure 5 caption was updated to specify that it is a global analysis and the figure is updated as follows. The point about the cut-off was clarified in the comment above and homogenized with the rest of the analysis (cutoff at 1990).

Extended data Fig. 5

Figure comments:

Figure 1:

Missing panel labels. Pump and CPR data cannot be compared to sediment samples (ForCenS) because the pump/CPR data samples the surface ocean whereas the latter is a time integration of samples, sometimes significantly long, depending on the sedimentation rate. It might be appropriate to compare the sediment trap samples with the seafloor samples.

Reply: Please see our response to the comment above.

Figure 2:

I don't understand this figure. Is the width of the colored bands related to abundances? Why is before and after 1997 chosen for the cutoff? Is the transition to deeper water for the NS species driven by temperature or perhaps food availability? For example, crassiformis deepening by 45 meters because it was found in one location before and after 1997 at different depths? Were the conditions the same? Or did it migrate because hydrographic conditions changed its preferred thermal tolerance?

Reply: The width is indicating the recurrence of the depth at the maximum abundance per species. The cut-off was not chosen arbitrarily. The multinet samples, unfortunately, do not offer a broad temporal resolution, and 1997 serves as a practical cutoff point where we have sufficient data both before and after, enabling a meaningful comparison.

Concerning *G. crassaformis*, the species is identified in multiple locations before and after 1997 within the latitudinal band spanning 0 to 30°N of the Atlantic Ocean, as illustrated in the figure

R2. *G. crassaformis* is notably characterized by its deep-dwelling nature and absence of symbionts and here, exhibits a deepening of its habitat. This shift in depth is likely a response to hydrographic conditions that have altered its preferred thermal tolerance but without more environmental data, we cannot provide more precisions. The comparison takes into account the presence of the species in different locations and aims to elucidate the factors influencing its depth distribution over time.

In the revised version we added this clarification about *G. crassaformis* depth habitat deepening:

“At low latitudes, only three species appear to descend to greater depths over the past decades, the most robust signal being the 45 ± 21 m vertical deepening, between samples taken before and after 1997, for the symbiont-barren species *G. crassaformis* (Fig. 2A). This notable shift in dwelling depth may be attributed to changes in hydrological and ecological conditions, including alterations in water temperature or variations in food availability, which likely prompted their appearance in deeper waters.”, in lines 149-154.

Fig. R2: Number of observations per species in the low (from 0 to 30°N) and mid (30 to 50°N) latitudes, collected with plankton nets during spring and summer from the North Atlantic and Arctic Oceans, between 0 to 200 m water depth.

Figure 3: I have some concerns that there is significant sampling bias in these results. For example, the data from the 1940-1950 is from one location off the coast of N. America. With each decade, the number of locations sampled increased significantly, and thus do the normalized number concentration per cubic meter go down simply because the sampled area went up?

Reply: Yes, we fully take into account that the sampling distribution is a bias that could affect our datasets, and this is specifically addressed in the text (lines 185-192). This is the main reason we concentrate on the North Atlantic regions in most of the analyses to get more robust statistics. This was also the reason why for the depth habitat analysis, we focused only on sampling casts ranging same depth intervals (Fig. 2).

For the figure 3, we addressed this specific point, by quantifying the decreasing abundance trend with and without the data from 1940-1950, as explained in the main text in lines 185-192: **“Our data reveal a gradual decrease in surface and subsurface PF abundance across the different latitudinal bands in the North Atlantic Ocean, over the past decades, statistically significant between 0 and 50°N (Fig. 3). This abundance decrease is most pronounced in low to mid latitude regions (5.5 ± 0.05 (1SD) % for 0 to 30°N, and 24.24 ± 0.11 % for 30 to 50°N, between 1950 and 2018) (Figs. 3A, B; Extended Data Fig. 8), where the decline in abundance is particularly acute for subtropical and temperate species. These trends are even stronger when including the sparse data prior to 1950 in the low latitudes, (0 to 30°N), with an abundance decline reaching 42.08 ± 0.15 % from 1940 to 2018.”**

Figure 4: There is not enough detail in the paper, the figures, or the figure captions for me to fully grasp the findings presented in this figure.

Reply: We agree that the precedent version was somewhat complex. To simplify the figure 4, we split it into 2 separate figures (Figure 4 and Extended Data Figure 9), we added details in the figure captions, and we clarified parts of the main text (Lines 217-245):

“Species-specific ecological niches of marine biota are undergoing worldwide shifts due to warming and ocean acidification, partly resulting from anthropogenic CO₂ emissions^{5,45}. We use the saturation state of seawater with respect to calcite (Ω_{calcite}), as a metric for ocean acidification. Examining prevalent PF species from three latitudinal North Atlantic range bands, we find that their current habitats cover temperature and Ω_{calcite} ranges that largely align with, or exceed, projected mid-to-end-century changes under a realistic, “middle-of-the-road” scenario (SSP 2 – 4.5) (Extended Data Fig. 9). Notably, species such as *G. bulloides* exist today even in areas where Ω_{calcite} is below 1, meaning that dissolution should be favoured for calcite (Extended Data Fig. 9). In the tropics (Extended data Fig. 9 A, and D), at the end of the century, PF would be exposed to a combination of temperature and saturation states that is unlike anything experienced by any other PF species today (Fig. 4). In temperate and high-latitude areas, PF are expected to reach the approximate limit of their currently habitable temperature and saturation state range by 2100. A broader analysis indicates PFs are rather eurythermal and support a wide range of temperature, yet with species-specific temperature optima and distribution patterns (Extended Data Fig. 5).

Visualizing ecological niches in a 2-dimensional (Ω_{calcite} and temperature) space reflecting environmental conditions, future migration patterns can be projected (Fig. 4). The distribution of Ω_{calcite} and temperature at the time and location of sampling for every PF occurrence within the FORCIS represents the current PF ecological niche, plotted in grey in Fig. 4 for three different latitudinal bins. Each PF species occupies a subset of the current PF ecological niche, with some species adapted to warm, high- Ω_{calcite} waters (e.g. *G. ruber*), and some more familiar with cold, low- Ω_{calcite} waters (e.g. *N. pachyderma*). Using model predictions of future temperature and Ω_{calcite} , under the SSP 2 – 4.5 scenario, the future trajectory of these regional niches in terms of temperature and Ω_{calcite} can be predicted. Contrary to common perception, it is tropical, and not polar environments which will likely see the most important changes, despite polar amplification of climate change. By 2050, environmental conditions in most tropical locations will fall outside that of current PF-ecological niches. In temperate and polar zones too, current PF niches will move towards the limits of modern communities, but will largely remain within the temperature and Ω_{calcite} conditions occupied globally by PF.”

The revised figure 4:

Figure 4. Panels (A–D) show the modelled ocean temperature and Ω_{calcite} at collection time and collection water depth from the IPSL IPSL-CM6A-LR model output for all FORCIS Foraminifera locations globally (grey points). Marker species are shown in coloured points with *G. ruber* (blue), *T. sacculifer* (dark purple), and *G. glutinata* (green), in the tropics (A), *G. ruber* (blue), *T. sacculifer* (dark purple), *G. glutinata* (green), and *G. bulloides* (yellow), in the mid latitudes (B) and *G. glutinata* (green), *G. bulloides* (yellow), and *N. pachyderma* (orange) in the high latitudes (C). Also shown on each panel are the projected future conditions of all current niches of Foraminifera marker species in each latitudinal band in 2050 (purple) and 2100 (pink) based on the model output SSP 2 – 4.5. The pink and purple distributions show that future ocean environments will be in temperature and saturation conditions that are not currently experienced by any living Foraminifera species. Panel D shows a cartoon schematic of ocean temperature and saturation state conditions in the current niches inhabited by Foraminifera. The grey balloons encircle ocean show actual conditions. The purple balloon shows historical Foraminifera localities and their conditions projected to 2050, the pink balloon shows these locations projected to 2100. The only Atlantic group which finds itself outside of known modern conditions is the tropical biota who will (if they remain tropical) face unprecedented combinations of temperature and saturation state.

Minor note: The Fox paper erroneously used a thinly calcified (likely live at the time of collection) *N. dutertrei* and thus the 74% thinner modern specimen is simply due to comparing two different ontogenic stages (a live vs. dead foram), not OA.

Reply: This is an interesting note, and we do agree that the sampling approach used in the Fox paper was very limited with a total number of Foraminifera analyses of 4 shells of *N. dutertrei* and 4 specimens of *G. ruber*. Thus, we removed the reference from the text.

Referee #3

The manuscript titled 'Modern planktonic Foraminifera: migrating is not enough' by Chaabane et al., is an exceptional and highly topical piece of work. The manuscript is elegantly written with a broader audience in mind and therefore will be a cornerstone for future research and public discussions.

For the first time, horizontal and vertical distribution patterns were plotted using the invaluable dataset from ForCenS global census. The authors used this long-term dataset in a novel way to understand how planktonic foraminifer have migrated through space and time. The authors also took it one step further and projected possible migration patterns considering the two most prominent physico-chemical parameters of climate change influencing marine calcifiers (calcite saturation and temperature). This work provides an unprecedented look at the nuanced adaptive strategies of a vital ecosystem engineer that will have a significant impact on climate regulation into the future.

The methods and interpretations of the data are appropriate and robust. The figures are synthesised and presented in an engaging way with ample explanation in the extended data section. The caveats, particularly the number of observations, are mentioned and dealt with honestly. There are only a few typos and minor edits that need to be corrected (e.g., Sup Table 4 *G. hexagona* p values should have an *).

The quality of this global study is outstanding and its' original conclusions will have a far reaching influence beyond the marine micropaleontological community, therefore, I highly recommend this work be published in Nature.

Authors: We thank Reviewer #3 for the encouraging feedback and recommendation for publication.

Reply: We thank Reviewer #3 for the reading and we have carefully cross-checked the manuscript to avoid any typos. Supplementary table 4, *G. hexagona* is corrected.

Reviewer Reports on the First Revision:

Referees' comments:

Referee #1 (Remarks to the Author):

Modern planktonic Foraminifera: migrating is not enough
By Sonia Chaabane and others

This is my second review of the manuscript Modern planktonic Foraminifera: migrating is not enough by Chaabane and others. The manuscript has greatly improved, in general the text is much clearer, the methods are well described, the structure and flow of the text are now easier to follow. All my previous comments have been addressed.

As previously stated, this manuscript builds on Chaabane et al. (2023) Scientific Data where the FORCIS database was published. The ability to examine how plankton communities have changed vertically, spatially and through time is particularly novel and will be of interest to a wide audience, including marine biologists and paleontologists interested in the macro and micro fauna and flora, and ocean and climate modellers.

There are several significant findings, including the change in depth habitat and the declining abundances, particularly of many tropical/subtropical species. This work significantly contributes to knowledge of the changing biology of the oceans in response to warming. I found the section on the decline in planktonic foram abundance (starting line 181) to be particularly significant and only possible with the analyses of these large global datasets.

All the additions to the text in red significantly improve the manuscript, though I found the text still needs some revisions for clarity (outlined below).

Line 63-66: "These effects are exacerbated for organisms producing a calcium carbonate shell or skeleton, as acidification impedes calcification faster than warming favours it⁷, due to processes such as remineralization of the organic matter in the upper water column^{8,9}". The remineralization of organic matter is temperature dependent, but a different process and unrelated to acidification (because of rising CO₂). Also organic matter (i.e. the availability of food), impacts all organisms regardless of whether they have a carbonate shell. I think this can be resolved by rewording and making these two separate sentences.

Line 133: reword to remove the double negative of "not to be unrelated"

Lines 148-160: I found this section very difficult to follow. It refers to five species in the temperature North Atlantic, but then only three species are listed: dutertrei, inflata and universa. These are all referred to as thermocline-dwelling, but according to Supplementary Table 1, O. universa is spinose

symbiotic, so probably mixed-layer rather than thermocline-dwelling. This section is also at odds with the abstract which states “some symbiont-barren species descending in the water column” but includes *O. universa* which has symbionts. The text goes on to say that 6 out of 8 species show no significant vertical migration in mid-latitudes, so does that mean that two species do have significant vertical migration in mid-latitudes? If so, which ones? I like the quantification e.g., 45 ± 21 m of the vertical deepening, as different species occupy different depth habitats, it would be useful to also state the broader range of depths, e.g. *crassaformis* deepens by 45 ± 21 m, from ~55 to 100 m.

Lines 175-179: This is a long sentence.

Line 195: I really like this section. *G. siphonifera* is highlighted as a 80% decline in abundance, but on Figure 3 it appears there are several other species that also significantly decline, e.g. *sacculifer*, *universa* and I’m not sure why *siphonifera* alone is listed in the text.

Supplementary Table 1: As in my previous review, I still don’t know and it is not stated what are the empty/blank spaces in this table in the photo-symbiont column. I think it means no data, please clarify.

Supplementary Table 3 also has blank spaces, please clarify if blank is different to “-“

Line 206: change *G. uvula’s* to *G. uvula*

Throughout they are termed planktonic Foraminifera (lower case p and upper case F). Why is this? It seems inconsistent with other papers. Brummer and Kucera (2022) p.30 states “we recommend the following usage: capitalized, the word “Foraminifera” is Latin and refers to a taxonomic unit; not capitalized, the word is vernacular and English and can be both singular and plural (like “sheep”)”. I suggest using planktonic foraminifera (no capitalization) in line with major works by the community over the last 60 years and the ForCenS database.

I found the final part of the abstract a bit too vague “has broader implications for the evolution of marine life under multiple stressors” and think that can be strengthened.

Abstract: I suggest removing the start of the opening sentence “Anthropogenic activities, in particular” so it starts “Rising CO₂ emissions, provoke ocean warming...”. Note this is not because I disagree that rising CO₂ emissions are anthropogenic, I just think it is smoother that way.

Line 328: insert ‘the’ between over and last century.

Referee #2 (Remarks to the Author):

The authors utilize the recently developed FORCIS database and the preexisting ForCenS databases to

assess how anthropogenically driven climate change has already altered the distribution and diversity of planktic foraminifera both vertically within the water column and latitudinally with a focus on the North Atlantic. The FORCIS database, which compiles a wide range of planktic foraminifera data and sampling techniques, is used here in a novel way to quantify impacts of modern climate change on planktic foraminifera communities. The sheer amount of data the authors are wrangling with, and the questions being asked of it are especially exciting. Results from these analyses will ultimately be applicable to, and potentially transformative for, a wide range of disciplines.

As written, I broadly agree with the comments of Reviewer 2, which I find to have been partially but not entirely satisfied in the current revision. In particular, I am concerned about sampling biases within and between the sample types used for these analyses. In most cases the influence of these biases could actually be probed with this database. Moreover, the methods still require substantially more detail. I have made several comments below, most of which touch on these themes, and that I hope the authors will find useful.

1. The comparability of biodiversity across different sample types needs further validation. The authors state they focused on presence/absence rather than quantitative flux. However, this does not necessarily address the fundamental concern raised by Reviewer 2. Cumulative flux as captured in sediments, and to some extent sediment traps, would be expected to sample greater diversity than any snapshot method. This is actually born out in Figure 1, where it appears that sediment traps sample overall greater diversity than the other “modern” methods at the same latitudes. It may be that diversity assessed from these various sampling methods are just not all directly comparable. I suggest the authors could better probe these biases by actually comparing diversity between sed traps and plankton tow/pump/CPR in the same regions through the same sampling period. This should be possible using this dataset and would be an analysis that would alone contribute to the field.

2. In their response to Reviewer 2, the authors point to a single paper from the Southern Ocean to justify direct comparability between CPR and plankton tows. However, Meiland et al. (2016) only addresses the near subsurface assemblage and explicitly discusses the problem of CPR under sampling planktic foraminifera. It simply does not make sense that the diversity captured in the upper 10(s) of meters would be equivalent to that over the upper 100s of m of the water column. A nice example of this can actually be found in Figure 2 of this submission, which includes several species which were abundant within the water column but not collected at all in the upper 10s of m during one or all net tows. How would these be captured by CPR? As above, this is a bias that could be appropriately and robustly tested within the FORCIS database.

3. In addition it would be worth evaluating whether reliance on different sampling methods (or on different species concepts) through time might impact the apparent reduction in biodiversity in the dataset. This could potentially be explored with an additional panel in Figure 1.

4. Despite the focus on the North Atlantic, this region still has sparse data for interpreting depth migrations such that I wonder how seasonal and/or regional changes in depth habitat can be ruled out. I

spent some time trying to filter this dataset as described by the authors and found that the depth habitat comparison seems to rely on a relatively small number of sampling campaigns. This left me concerned that the reported number of observations may be unintentionally masking a lack of coverage across both time and space. For example, I could find only one pre-1997 (1996) sampling campaign between 0-30 degrees that would meet the specifications of the authors. While many samples were taken on this cruise, this effort occurred entirely in spring (~April) in comparison to the post-1997 campaigns which sampled primarily in summer, with a few spring cruises. How might the authors rule out a seasonal signal in this data? Greater transparency about the timing and location (i.e. depth habitats within and outside of the North Atlantic gyre are probably not comparable) of each sample is warranted. I'm not able to reproduce the number of observations reported for the 30-50 campaigns so cannot comment on these results.

There is another minor critique embedded here: I admittedly spent an miniscule fraction of time compared to the authors with this database, and it is entirely possible I missed something in my attempt to quickly replicate this analysis. However, my inability to efficiently find all of the samples included in these analyses speaks to a lack of clarity in the methods that should be addressed.

Minor comments

44: define acronym at first use

63-66: could you clarify the connection between remineralization and the first half of the sentence?

68-70: please clarify this sentence as well

133: I believe "not to be unrelated" should just be "unrelated"

117-118: "between modern and Holocene" should be rewritten as the modern is technically also Holocene. Generally I'd recommend consistent terminology for data from the FORCIS vs ForCenS dataset. Pre- and post- industrial are the most frequently used in this paper, so I might use these throughout

205: remove "in". Also note that the study cited here models specifically a non-spinose *and* symbiont barren species, which may be meaningful in this context.

264-270: These lines seem to conflate a laboratory induced decalcification of a calcifying taxa with a distant lineage of foraminifera that do not calcify at all. I think this is unintentional, but it currently reads as a misunderstanding of the differences between adaptation and evolution, and should be rewritten.

321: Please expand on how different mesh sizes were normalized. This likely has a bearing on the sampled diversity

399-403, How do abundances through time relate to forams/m³ (Figure 3)? Could the authors provide a more detailed explanation?

404-408: Can you add more clarity about how the regression portion of the ANOVA was done?

Figures: Figure labels and titles should be made clearer throughout

Figures: If allowable by the journal, please use a larger font sizes on the heatmaps

Figure 1: Please increase opacity in the key so colors are more visible

Figure 2: Can you indicate here which species are the major species?

Figure 2: Here and throughout, note that 30-50 degrees is not generally considered Arctic

Figure 3: Please make sure brackets are correct here and in the extended data figures

Fig 2 & 3 have the same colors for different parameters. Could you differentiate these for clarity?

Ext Data Fig 3: check the spelling of pre- and post- industrial selling

Ext Data 5: What does number of profiles mean?

Author Rebuttals to First Revision:

15-07-2024

COMMENTS TO THE AUTHORS OF NATURE MANUSCRIPT 2023-10-19099A (“MODERN PLANKTONIC FORAMINIFERA: MIGRATING IS NOT ENOUGH”) WITH POINT-BY-POINT RESPONSES FROM THE AUTHORS (IN BLUE)

SECOND ROUND OF REVIEW

Your manuscript, "Modern planktonic Foraminifera: migrating is not enough", has now been seen by 2 referees. The previous Referee # 2 could not review in this round, so we found a replacement referee with similar expertise. You will see from their comments below that while they find your work of interest, some important points are raised. We are interested in the possibility of publishing your study in Nature, but would like to consider your response to these concerns in the form of a revised manuscript before we make a final decision on publication.

We therefore invite you to revise your manuscript taking into account the following points.

Referee # 2 feels that the diversity assessed from the different sampling methods in different sampling periods cannot be directly comparable. As such, these potential biases should be addressed in your analysis and discussed in the manuscript, along with other referee concerns related to data availability and clarity in the methods' description (Referee # 2). We feel that the suggestions of Referee # 2 would further increase the reproducibility of the paper.

Authors: We would like to thank the editors and the two anonymous reviewers for their time and supportive feedback on our manuscript. Based on the reviewers' comments, we have revised the text, analysis, figures, tables, and supplementary material.

Below, you will find our detailed responses to the reviewers' comments. We have addressed each comment meticulously in both our responses and the manuscript, including text and figures where necessary. We hope that this second revised version will meet the criteria for publication in Nature.

Referee #1

Modern planktonic Foraminifera: migrating is not enough
By Sonia Chaabane and others

This is my second review of the manuscript Modern planktonic Foraminifera: migrating is not enough by Chaabane and others. The manuscript has greatly improved, in general the text is much clearer, the methods are well described, the structure and flow of the text are now easier to follow. All my previous comments have been addressed.

As previously stated, this manuscript builds on Chaabane et al. (2023) Scientific Data where the FORCIS database was published. The ability to examine how plankton communities have changed vertically, spatially and through time is particularly novel and will be of interest to a wide audience, including marine biologists and paleontologists interested in the macro and micro fauna and flora, and ocean and climate modellers.

There are several significant findings, including the change in depth habitat and the declining abundances, particularly of many tropical/subtropical species. This work significantly contributes to knowledge of the changing biology of the oceans in response to warming. I found the section on the decline in planktonic foram abundance (starting line 181) to be particularly significant and only possible with the analyses of these large global datasets.

Authors: We would like to thank the reviewer #1 for the insightful review on our manuscript. Their feedback has been invaluable in enhancing our work.

All the additions to the text in red significantly improve the manuscript, though I found the text still needs some revisions for clarity (outlined below).

Line 63-66: “These effects are exacerbated for organisms producing a calcium carbonate shell or skeleton, as acidification impedes calcification faster than warming favours it⁷, due to processes such as remineralization of the organic matter in the upper water column^{8,9}”. The remineralization of organic matter is temperature dependent, but a different process and unrelated to acidification (because of rising CO₂). Also organic matter (i.e. the availability of food), impacts all organisms regardless of whether they have a carbonate shell. I think this can be resolved by rewording and making these two separate sentences.

Reply: Thank you for your insightful comments. We have revised the sentence to separate the discussion on acidification and remineralization, clarifying that remineralization, while temperature-dependent, impacts the availability of food for most organisms:

“These effects are particularly severe for organisms producing a calcium carbonate shell or skeleton, as acidification impedes calcification faster than warming favours it⁷. Furthermore, the increasing remineralization of organic matter in the upper water column^{8,9}, in response to oceanic warming, could enhance the nutrient availability”, in lines 64-68.

Line 133: reword to remove the double negative of “not to be unrelated”

Reply: Corrected in line 136.

Lines 148-160: I found this section very difficult to follow. It refers to five species in the temperature North Atlantic, but then only three species are listed: dutertrei, inflata and universa. These are all referred to as thermocline-dwelling, but according to Supplementary Table 1, O. universa is spinose symbiotic, so probably mixed-layer rather than thermocline-dwelling. This section is also at odds with the abstract which states “some symbiont-barren species descending in the water column” but includes O. universa which has symbionts. The text goes on to say that 6 out of 8 species show no significant vertical migration in mid-latitudes, so does that mean that two species do have significant vertical migration in mid-latitudes? If so, which ones? I like the quantification e.g., 45±21 m of the vertical deepening, as different species occupy different depth habitats, it would be useful to also state the broader range of depths, e.g. crassaformis deepens by 45±21 m, from ~55 to 100 m.

Reply: We thank the reviewer for pointing this out. In the revised version, we tried to make this paragraph clearer and add the depth range of the species that show a deepening of the depth habitat. We also deleted symbiont-barren from the abstract in line 45.

“A unique feature of the FORCIS database is the possibility to track vertical changes in the PF distribution at the species level over time. At low latitudes, two species significantly descended to greater depths over the past few decades. The symbiont bearing *G. calida* exhibits a significant deepening of their habitat depth by 53.5 ± 15 m (from ~20 to 75 m). The symbiont-barren species *G. crassaformis* descended by 45 ± 21 m (from ~55 to 100 m), between samples taken before and after 1997 (Fig. 2A). This notable shift in depth habitat may be attributed to changes in hydrological and ecological conditions, including alterations in water temperature or variations in food availability, which likely prompted their appearance in deeper waters. In the mid-latitudes, symbiont-barren species such as *N. dutertrei* (20 ± 5 m deepening before and after 1997; from ~50 to 70 m), *G. inflata* (40 ± 5 m from ~30 to 70 m), and *N. incompta* (40 ± 4 from ~10 to 50 m) significantly descended in the thermocline and mixed layer (Fig. 2B). Only two out of eight symbiont bearing PF species show significant vertical migration, i.e., *G. siphonifera* (20 ± 5 m from ~50 to 70 m) and *O. universa* (40 ± 3 m from ~10 to 50). Based on these observations, the trophic regime of PF species appears to constrain the vertical distribution changes of species in the North Atlantic.” in lines 151-165.

Lines 175-179: This is a long sentence.

Reply: Thank you for your valuable feedback. We have revised and divided the sentence to clearly separate the discussion of vertical migration and isotherm deepening:

“This suggests that vertical migration is influenced by the trophic regime of each species. However, this migration is slower than the deepening of isotherms, which was estimated at -6.6 ± 18.8 m/decade between 1980 and 2015. This rate is predicted to accelerate to -32 m/decade by the end of the century under a high emission scenario³⁷. We speculate that this will limit the ability of PF to cope with these warmer environments.”, in lines 180-185.

Line 195: I really like this section. *G. siphonifera* is highlighted as a 80% decline in abundance, but on Figure 3 it appears there are several other species that also significantly decline, e.g. *sacculifer*, *universa* and I’m not sure why *siphonifera* alone is listed in the text.

Reply: We did not discuss all species to avoid making the text too long and only gave an example of the species for which the abundance declined the most. In the revised version of the manuscript, we clarify this: “Even though early (pre-1960s) census data are rare, the post-1960s data show a statistically significant decline in abundance for 14 out of 26 species (e.g. up to a maximum decline of 80 ± 0.3 % between 1950 and 2010 for the subtropical species *G. siphonifera*).”, lines 198-201.

Supplementary Table 1: As in my previous review, I still don’t know and it is not stated what are the empty/blank spaces in this table in the photo-symbiont column. I think it means no data, please clarify.

Reply: We added below the Supplementary Table 1 “No data (-)” and “-” to the empty/blank spaces.

Supplementary Table 3 also has blank spaces, please clarify if blank is different to “-“

Reply: We added below the Supplementary Table 3 “No data (-)” and “-” to the empty/blank spaces.

Line 206: change *G. uvula*’s to *G. uvula*

Reply: Corrected in line 213.

Throughout they are termed planktonic Foraminifera (lower case p and upper case F). Why is this? It seems inconsistent with other papers. Brummer and Kucera (2022) p.30 states “we recommend the following usage: capitalized, the word “Foraminifera” is Latin and refers to a taxonomic unit; not capitalized, the word is vernacular and English and can be both singular and plural (like “sheep”)”. I suggest using planktonic foraminifera (no capitalization) in line with major works by the community over the last 60 years and the ForCenS database.

Reply: We use “Foraminifera” in the context of formal zoological nomenclature as a name for a taxon according to the ICZN (Ride, 1999). We agree with the reviewer that this is inconsistent with other papers. However, the term “planktonic Foraminifera” was agreed upon by the FORCIS working groups and is used in all FORCIS papers.

I found the final part of the abstract a bit too vague “has broader implications for the evolution of marine life under multiple stressors” and think that can be strengthened.

Reply: Thank you for your input. We have strengthened the final part of the abstract in the revised version of the abstract:

“Our insights into the adaptation of planktonic Foraminifera during the Anthropocene reveal that 'migration is not enough' to ensure survival. This underscores the urgent need to understand how the interplay of climate change, ocean acidification, and other stressors will impact the survivability of large parts of the marine realm.”, in lines 53-56.

Abstract: I suggest removing the start of the opening sentence “Anthropogenic activities, in particular” so it starts “Rising CO2 emissions, provoke ocean warming...”. Note this is not because I disagree that rising CO2 emissions are anthropogenic, I just think it is smoother that way.

Reply: We deleted “Anthropogenic activities, in particular” from the abstract in the revised version in line 40.

Line 328: insert ‘the’ between over and last century.

Reply: We added “the” in line 338.

Referee #2

The authors utilize the recently developed FORCIS database and the preexisting ForCenS databases to assess how anthropogenically driven climate change has already altered the distribution and diversity of planktic foraminifera both vertically within the water column and latitudinally with a focus on the North Atlantic. The FORCIS database, which compiles a wide range of planktic foraminifera data and sampling techniques, is used here in a novel way to quantify impacts of modern climate change on planktic foraminifera communities. The sheer amount of data the authors are wrangling with, and the questions being asked of it are especially exciting. Results from these analyses will ultimately be applicable to, and potentially transformative for, a wide range of disciplines.

Authors: We would like to thank the reviewer #2 for the helpful comments and suggestions. We sincerely appreciate the constructive feedback provided.

As written, I broadly agree with the comments of Reviewer 2, which I find to have been partially but not entirely satisfied in the current revision. In particular, I am concerned about sampling biases within and between the sample types used for these analyses. In most cases the influence of these biases could actually be probed with this database. Moreover, the methods still require substantially more detail. I have made several comments below, most of which touch on these themes, and that I hope the authors will find useful.

1. The comparability of biodiversity across different sample types needs further validation. The authors state they focused on presence/absence rather than quantitative flux. However, this does not necessarily address the fundamental concern raised by Reviewer 2. Cumulative flux as captured in sediments, and to some extent sediment traps, would be expected to sample greater diversity than any snapshot method. This is actually born out in Figure 1, where it appears that sediment traps sample overall greater diversity than the other “modern” methods at the same latitudes. It may be that diversity assessed from these various sampling methods are just not all directly comparable. I suggest the authors could better probe these biases by actually comparing diversity between sed traps and plankton tow/pump/CPR in the same regions through the same sampling period. This should be possible using this dataset and would be an analysis that would alone contribute to the field.

Reply: Thank you for the insightful comments. We agree with the reviewer that the plankton net and sediment trap data comparison could induce confusion because of inherent collection biases from different devices. This is briefly mentioned in the text in lines 115 to 118.

The plankton net profiles comprise many samples collected at the same time, with the same device, and from the same coordinates. The sediment trap samples were collected over an average time span of 15 days and are primarily from depths greater than 3000 meters in our database. In FORCIS, the whole time series obtained from each sediment trap is represented by its profile_id (Chaabane et al. 2023 Scientific Data).

In Fig. 1, in most cases the diversity derived from the plankton net profile exceeds the ones obtained from the sediment trap samples (yellow vs. orange dots). In few cases it is the contrary for example, in the southern hemisphere and northern mid-latitudes, certain sediment trap samples exhibit slightly greater diversity when compared to some plankton tow samples. We attribute this discrepancy primarily to the limited sampling effort in these regions with plankton

nets, which cannot cover all seasons comprehensively, whereas typically plankton nets find great diversity due to better sample preservation.

To address the reviewer's comment, we conducted an analysis comparing diversity between sediment traps and plankton tow samples within identical latitude and longitude bins ($2^\circ \times 2^\circ$), sampling year, and month. Despite limited data (4 sediment traps samples vs. plankton net; red dots in figure R1), the results show that species richness obtained from sediment traps is lower (by 1 species only) though comparable to that from plankton nets.

Fig. R1. Boxplot of residual species richness calculated from the plankton net profiles and the sediment traps samples (species richness in plankton net profiles - species richness in trap samples) at the same location (2° latitude and 2° longitude) and time (same year and month). Individual data points are shown as red dots with slight horizontal jitter to prevent overlap, where each dot represents a unique observation of residual species richness.

This suggests that our methodology in comparing profiles for the plankton net data and samples for the sediment traps data is robust. The sinking time for a Foraminifera from the surface to > 3000 m from 3 to 10 days (Takahashi and Bé, 1984), therefore, we would expect a sediment-trap with a 15 days collection period, if no strong dissolution occurs in the water column, to account for most of the Foraminifera species caught in a multinet profile at a similar time and place

Finally, we would like to stress that in this section, our intention is to highlight differences in species diversity between past and modern data, regardless of the sampling device used. We do not aim to compare different modern collection techniques.

2. In their response to Reviewer 2, the authors point to a single paper from the Southern Ocean to justify direct comparability between CPR and plankton tows. However, Meilland et al. (2016) only addresses the near subsurface assemblage and explicitly discusses the problem of CPR under sampling planktic foraminifera. It simply does not make sense that the diversity captured in the upper 10(s) of meters would be equivalent to that over the upper 100s of m of the water

column. A nice example of this can actually be found in Figure 2 of this submission, which includes several species which were abundant within the water column but not collected at all in the upper 10s of m during one or all net tows. How would these be captured by CPR? As above, this is a bias that could be appropriately and robustly tested within the FORCIS database.

Reply: We thank the reviewer for this comment. We agree with the reviewer that the CPR is unable to sample deep-dwelling organisms and small specimens. However, many studies such as that of McQuatters-Gollop et al. (2010) and Hosie et al. (2014) found that the CPR captures consistent and comparable changes in plankton distribution and community composition over time, even for small size plankton.

The CPR diversity data presented in Figure 1 originates from the CPR deployed in the Southern Ocean where fewer species naturally occur. The device occasionally captured up to five species simultaneously, with an average of one to three species per tow.

Unfortunately, the FORCIS database does not have samples available for the same year, month, and coordinates for both plankton net and CPR that would allow a direct comparison. Our main message is that biodiversity remains unchanged with or without the CPR data. The observed pattern of increasing species richness in the high latitudes, as calculated from the FORCIS database, is robust and reliable.

We are reluctant to completely remove the CPR data from our study, because it represents high resolution and good spatial coverage in areas of the ocean where (barely) no other data exist. Thus, even though we note that its inclusion into our formal analysis makes no difference to our results, we prefer to discuss its contribution to our understanding of PF change on a qualitative level, and posit that these data may become more viable for quantitative studies in future.

To further refine our analysis and address the reviewer's concern, we added to the supplementary material a figure comparing the profiles of plankton net data (represented by yellow dots) to the sediment trap samples (represented by orange dots) and we kept the former figure 1 to allow the readers to distinguish between the outcomes of the different sampling methods which is pointed to in the main text in lines 116 to 119. Overall, even after excluding the CPR and pump data from the analysis in figure S1 the main result of the increase of the diversity in the high latitudes is unchanged.

Fig. S1

Figure S1. Diversity Changes in Planktonic Foraminifera: (A) Map of diversity of planktonic Foraminifera (number of species using the compiled taxonomy4), and (B) Comparison of pre-industrial diversity inferred from surface sediment ForCenS database with living planktonic Foraminifera FORCIS database over the last 100 years fitted by a Generalized Additive Model (GAM) smoothing curve at the 95th percentile of species richness at each 10° latitude bin (grey line: ForCenS data; red line: FORCIS data). The number of total observations in FORCIS and ForCenS together for each 5-degree latitudinal bin and each species richness level are presented in the histograms.

3. In addition it would be worth evaluating whether reliance on different sampling methods (or on different species concepts) through time might impact the apparent reduction in biodiversity in the dataset. This could potentially be explored with an additional panel in Figure 1.

Reply: We thank the reviewer for the insightful suggestion. Our analysis spans the entire post-industrial period for FORCIS and relies on pre-industrial data from the ForCenS database. In Figure 1, combining all the data from 1910 to 2018 smooth the data and our diversity estimating (red fit vs grey one; Fig. 1) is based on spatially gridding the database into 10° latitudinal bands and using the 95th percentile of species richness for all devices. We then fitted with a Generalized Additive Model (GAM) to capture the variability in species richness along the latitudinal gradient. With this approach we believe that these metrics will estimate the minimal diversity in each latitudinal band.

To further clarify the different sampling techniques used in our dataset, we already added in Fig.1 histograms that describe the number of total observations in FORCIS and ForCenS together for each 5-degree latitudinal bin and each species richness level.

To address the reviewer comment, we added in the supplementary material a new figure (Fig. S2) to show species richness together with the number of observations obtained from the different devices through time. In figure S2, the diversity shows an increase over time resulting

from an increased sampling effort and leading to better spatial and temporal coverage after 1990. This shows that the spatial and temporal resolution of sample methods are not getting lower through time, in average, neither is the spatial or temporal coverage. Thus, if we see any diversity pattern change in the figure 1, it is real.

Fig. S2

Fig. S2. Number of observations and diversity Time Series. (A) Number of observations and (B) species richness time series the FORCIS database, based on Planktonic Foraminifera samples collected using four different sampling devices: plankton net (yellow dots), sediment traps (orange dots), CPR (purple dots), and pump (pink dots).

4. Despite the focus on the North Atlantic, this region still has sparse data for interpreting depth migrations such that I wonder how seasonal and/or regional changes in depth habitat can be ruled out. I spent some time trying to filter this dataset as described by the authors and found that the depth habitat comparison seems to rely on a relatively small number of sampling campaigns. This left me concerned that the reported number of observations may be unintentionally masking a lack of coverage across both time and space. For example, I could find only one pre-1997 (1996) sampling campaign between 0-30 degrees that would meet the specifications of the authors. While many samples were taken on this cruise, this effort occurred entirely in spring (~April) in comparison to the post-1997 campaigns which sampled primarily in summer, with a few spring cruises.

How might the authors rule out a seasonal signal in this data? Greater transparency about the timing and location (i.e. depth habitats within and outside of the North Atlantic gyre are probably not comparable) of each sample is warranted. I'm not able to reproduce the number of observations reported for the 30-50 campaigns so cannot comment on these results.

Reply: The temporal and spatial coverage in some areas is indeed low in the database. For that reason, we chose to focus on areas where the coverage is better, such as in the North Atlantic Ocean. The reviewer probably could not reproduce the analysis due to a lack of clarity in the text, which we corrected in the following. In the mid latitude of the North Atlantic Ocean, we filtered the dataset by latitude (30 to 50 degrees), depth (0 to 200 meters), and excluded samples collected using sediment traps, CPR, and pumps. We focused on samples collected during the summer and spring seasons. After grouping by profile ID and counting unique sample IDs, we retained profiles with four or more samples collected using a net sampling device. The number of profiles found is 59 and includes 259 unique depths at the maximum abundance of different species found in each profile.

We corrected that in the main revised text:

“It was assessed for two latitudinal bands from 0 to 30°N and from 30 to 50°N in the North Atlantic Ocean (Fig. 2). Only multinet data sampled across the upper 200 m and at profiles presenting higher or equal to four samples and the same sampling resolution (e.g., depth separations) during both spring and summer were selected. These data cover the period between 1980 and 2018. We focused on spring and summer due to increased biological productivity, warmer sea surface temperatures, and greater data availability from frequent research cruises during these seasons.” in lines 450-456.

The code to generate the analysis that the reviewer could not reproduce is posted on Zenodo: <https://zenodo.org/records/10881387> and will be available and linked to the publication.

This analysis is definitely a first step in assessing the depth patterns, and might still uncover some seasonality effects. FORCIS database has patchy data, which means the multinet samples are a bit sparse. Multinet towing is rather recent and was not done in all seasons. In the North Atlantic and Arctic Oceans most of the sampling occurred in spring and summer for meteorological reasons. To make the analysis on the depth habitat if we chose only spring, the amount of data was not sufficient. We therefore decided to select and use both spring and summer data.

In addition, there are other reasons to concentrate on these seasons:

- Seasonal Blooms: Spring and summer are typically associated with increased biological productivity and phytoplankton blooms in the North Atlantic and Arctic

Ocean. Planktonic Foraminifera populations often respond to these blooms by increasing in abundance as they feed on phytoplankton and organic matter.

- **Temperature and Stratification:** Spring and summer are characterized by warm sea surface temperatures and increased thermal stratification in the water column. These seasonal changes can influence the vertical distribution of planktonic foraminifera as they respond to variations in temperature and nutrient availability.

There is another minor critique embedded here: I admittedly spent an miniscule fraction of time compared to the authors with this database, and it is entirely possible I missed something in my attempt to quickly replicate this analysis. However, my inability to efficiently find all of the samples included in these analyses speaks to a lack of clarity in the methods that should be addressed.

Reply: We enhanced the quality of the text in the Material and Method section regarding the depth habitat to make it clearer. All codes for generating the figures are posted on Zenodo (<https://zenodo.org/records/10881387>).

“It was assessed for each latitudinal band from 0 to 30°N and from 30 to 50°N in the North Atlantic Ocean (Fig. 2). Only multinet data sampled across the upper 200 m and at profiles presenting higher or equal to four samples and the same sampling resolution (e.g., depth separations) during both spring and summer were selected. These data cover the period between 1980 and 2018. We focused on spring and summer due to increased biological productivity, warmer sea surface temperatures, and greater data availability from frequent research cruises during these seasons” in lines 450-456.

Minor comments

44: define acronym at first use

Reply: We define “PF” as “planktonic Foraminifera” in line 41.

63-66: could you clarify the connection between remineralization and the first half of the sentence?

Reply: In the revised version this sentence has change as follows:

“These effects are particularly severe for organisms producing a calcium carbonate shell or skeleton, as acidification impedes calcification faster than warming favours it⁷. Furthermore, the increasing remineralization of organic matter in the upper water column^{8,9}, in response to oceanic warming, could enhance the nutrient availability” in lines 64-68.

68-70: please clarify this sentence as well

Reply: We rephrased the sentence in the revised version to make it clearer:

“Comparable environmental crises have occurred in the geological past, but at much slower rates. For instance, significant surface ocean acidification was observed during the last deglaciation and the onset of the Holocene¹².” in lines 70-73.

133: I believe “not to be unrelated” should just be “unrelated”

Reply: Corrected in line 136.

117-118: “between modern and Holocene” should be rewritten as the modern is technically also Holocene. Generally I’d recommend consistent terminology for data from the FORCIS vs ForCenS dataset. Pre- and post- industrial are the most frequently used in this paper, so I might use these throughout

Reply: We agree with the reviewer and we changed “between modern and Holocene” by “**pre-industrial and post-industrial**” in the revised version in lines 120-121.

205: remove “in”. Also note that the study cited here models specifically a non-spinose *and* symbiont barren species, which may be meaningful in this context.

Reply: We deleted “in” from line 211. We also didn’t add “symbiont barren species” in this case because the species cited here are *G. uvula* and *G. glutinata*, and both are photo-symbionts facultative.

264-270: These lines seem to conflate a laboratory induced decalcification of a calcifying taxa with a distant lineage of foraminifera that do not calcify at all. I think this is unintentional, but it currently reads as a misunderstanding of the differences between adaptation and evolution, and should be rewritten.

Reply: We thank the reviewer for this suggestion. We rephrase the text to be clearer: **“A culture experiment of Evans and Erez⁴⁸ confirmed that two PF species, *G. ruber* and *G. siphonifera*, can survive post-shell dissolution, recalcify, and adapt to low pH conditions. Recalcification of dissolved foraminiferal tests has been validated in both field and laboratory conditions^{51–53}, suggesting that PF could live shell-less in low Ω_{calcite} regions as predicted based on future scenarios (Fig. 4).”**, in lines 270-275.

321: Please expand on how different mesh sizes were normalized. This likely has a bearing on the sampled diversity

Reply: We appreciate the reviewer's concern regarding potential biases in our data due to different mesh sizes used in sampling. The data extracted from plankton net and pump samples were standardized to ensure consistency across measurements. Specifically, we converted the abundance data obtained from FORCIS into individuals per cubic meter (individuals/m³). For coarse fractions sampled with mesh sizes greater than 100 μm , we applied a standardization method described in Chaabane et al. 2024 (in *Limnology and Oceanography: Methods*).

The potential effect of different mesh sizes in plankton nets on the estimate of the sampled diversity has been minimized in our analyses by taking only the presence/absence of species. With this approach, the patterns of diversity are less sensitive to sampling biases as for example the mesh size. In addition, we didn’t use the CPR data in this analysis since we focused mainly on the Northern Hemisphere where we do not have CPR species counts.

In the revised version we added more details about this normalization technique:

“Specifically, where coarse fractions were sampled using a mesh size greater than 100 μm , we employed the approach described in Chaabane et al. ⁶¹ for standardization. This

approach involves converting the abundance data extracted from different sampling devices from plankton net and pump into a common unit of individuals per cubic metre (individuals/m³). To achieve this, we used size-normalized catch model equations obtained from sampling depths on total planktonic Foraminifera of cytoplasm-filled and empty tests. By applying these equations, we were able to accurately quantify the abundance of coarse fractions down to 100 µm. This facilitates the computation of assemblages as it would have appeared if all material had been sampled using a 100 µm net, thus ensuring consistency across different sampling devices and conditions.”, in lines 326-335.

399-403, How do abundances through time relate to forams/m³ (Figure 3)? Could the authors provide a more detailed explanation?

Reply: In this section, we aimed to analyse the abundance of PF as a time series, specifically focusing on how the abundance varies per decade. The text in this section was modified as following:

“To determine the temporal variation in PF abundance, we analysed data of PF per cubic metre (#/m³) collected with plankton nets and pumps across different depths (0 to 200 metres) and geographical regions (from the North Atlantic to the Arctic Oceans). This data was then aggregated within three distinct latitudinal bands: 0 to 30°N, 30 to 50°N, and 50 to 90°N. For each decade since 1940, we normalized the total abundance of PF for each species within these bands. Normalization from 0 to 1 was performed to facilitate comparison across species and regions by standardizing the data, removing the effects of differing scales of abundance. This approach allows us to observe relative changes in abundance over time, making it easier to identify trends and patterns.” in line 410-418.

404-408: Can you add more clarity about how the regression portion of the ANOVA was done?

Reply: We added some clarification about the use of ANOVA in the revised version.

“To investigate whether there were significant changes in the abundance of different species within each latitudinal band, an analysis of variance (ANOVA) was conducted using the 'anova()' function on the fitted regression models. ANOVA helps in determining if the observed variations in species abundance across time bins are statistically significant.”, in lines 418-422.

Figures: Figure labels and titles should be made clearer throughout

Reply: We have tried to increase the sizes on the heatmaps when it is possible.

Figures: If allowable by the journal, please use a larger font sizes on the heatmaps

Reply: Done.

Figure 1: Please increase opacity in the key so colors are more visible

Reply: Done.

Figure 2: Can you indicate here which species are the major species?

Reply: All species in the figure 2 are major species. In table Supplementary Table 1 we highlighted the major species.

Figure 2: Here and throughout, note that 30-50 degrees is not generally considered Arctic

Reply: corrected in the caption of figure 2 and text in line 665.

Figure 3: Please make sure brackets are correct here and in the extended data figures

Reply: Done.

Fig 2 & 3 have the same colors for different parameters. Could you differentiate these for clarity?

Reply: Corrected

Ext Data Fig 3: check the spelling of pre- and post- industrial selling

Reply: Corrected.

Ext Data 5: What does number of profiles mean?

Reply: A profile contains all the samples collected during the same station, time, and cruise, which is now clarified in the caption of the extended data figure 5 as follow: **“The number of profile_id (contains all the samples collected during the same station, time, and cruise) observations at each 1°C-temperature bin is given in the upper panels.”** in lines 727-729.

Reviewer Reports on the Second Revision:

Referees' comments:

Referee #1 (Remarks to the Author):

This is my third review of the manuscript. This is a much improved and very much clearer manuscript. All my previous comments have been adequately addressed. The additional extended data figures are very welcome, and a great addition to the manuscript.

As previously stated, this is an extremely interesting and novel study, bringing together multiple data sets to determine recent and potential future plankton changes. It is widely accepted that anthropogenic warming will have an impact on life on Earth, but analysis and quantification of the effects that have already taken place over the last 50 or so years have been lacking. In this paper, the authors use FORCIS (a database of planktonic foraminifera with data going back to 1910). This manuscript builds on Chaabane et al. (2023) Scientific Data where the FORCIS database was published. The ability to examine how plankton communities have changed vertically, spatially and through time is particularly novel and will be of interest to a wide audience, including marine biologists and paleontologists interested in the macro and micro fauna and flora, and ocean and climate modellers.

Regarding Data and Methodology, and the validity of the approach, I still have one comment of confusion about the data handling and some minor edits where the text lacked some clarity.

Data and Methodology: I'm still unsure how some of the species/subspecies are treated in the dataset. For example it is *G. ruber ruber* on lines 139-140, 207, but only *G. ruber* line 143. Then line 343, *G. ruber* is separated into *G. ruber albus* and *G. ruber elongatus*, but with no mention of *G. ruber ruber*. This becomes very challenging when looking at northern displacement of occurrence (extended data Fig. 7) where *G. ruber* has a poleward migration, but *G. ruber ruber* has equatorward migration. I suggest that the authors lump and simply use *G. ruber* and explain in the text that this includes *G. ruber ruber*, *albus* and *elongatus*, and that the reporting in the data compilation prevents these species/subspecies being separated. This also has implications for the figures and tables. Alternatively, it needs to be stated in the text of how the distributions of *G. ruber*, differ from those of *G. ruber ruber*. Currently these species/subspecies have separate distributions in Fig. 2 and extended data fig. 8.

Minor edits

Line 52 abstract: the text states "this would radically alter low-latitude planktonic foraminifer communities", given that this is in the abstract, I think it should be more specific, in what way are the communities altered?

Line 68: change "enhance the" to "alter". The increase in nutrient availability would depend on the position in the water column.

Line 117: change “higher” to “greater” to avoid the combination of higher and high.

Line 230: reference needed for *G. bulloides* environment.

Line 412: change “This data was” to “These data were”

Line 450: “It was assessed” not clear what “it” is referring to.

Sometimes pre-industrial, sometime preindustrial

Referee #2 (Remarks to the Author):

I had reviewed this paper previously and find it substantially improved, especially with regards to explanations in the methodology. My reservations remain about the comparability of the collection methods included here, though I appreciate the steps taken to discuss these issues in the response comments.

I greatly appreciate the comparison provided between net and sediment trap data shared in the reviewer comments and find it heartening to see the similarity in diversity between the two. However, it is difficult to believe that this comparison wouldn't underestimate diversity from sediment traps. While the average collection period may only be 15 days, in practice few sediment traps are collecting only for 15 days in each location. Rather, many will be collecting for months or years. Thus, at a given location, a sediment trap will collect through a wider variety of seasons and conditions than a single tow at the same location, inevitably sampling more diversity.

This response also side steps the issue of the comparison to sediments (ForCens), in which case the comparison is a snapshot of species richness (plankton tows) compared to species richness in a sample aggregating over years, centuries, or longer. It should thus not be surprising that sediments would sample a greater diversity of species under most circumstances.

Finally, the addition of Fig S2 is wonderfully informative. However, I find this potentially less reassuring than the authors do. Fig S2 highlights not only substantial trends in the types of sampling available through time, but also the relatively low richness captured by CPR.

Minor comments:

54 – quotations not needed

159 – *N. dutertrei* has photosymbionts (Takagi et al., 2019 as cited, but also Bird et al., 2017 and others)

211 – As pointed out previously, this model specifically takes into account a non-spinose and symbiont bearing species. This is potentially quite significant in a calcification context as the presence of symbiosis should alter the immediate carbonate microenvironment regardless of spine presence. Moreover, both

the species mentioned are non-spinose, so by the logic of this sentence, shouldn't they also be impacted by ocean acidification rather than advantaged by their size? I wonder if something like Henehan et al., 2017 showing lower CI in smaller individuals might not better bolster the claim being made here.

329-332: Thank you for the additional detail here. Could you please provide or cite the equations used?

Author Rebuttals to Second Revision:

17-09-2024

COMMENTS TO THE AUTHORS OF NATURE MANUSCRIPT 2023-10-19099B (“MODERN PLANKTONIC FORAMINIFERA: MIGRATING IS NOT ENOUGH”) WITH POINT-BY-POINT RESPONSES FROM THE AUTHORS (IN BLUE)

THIRD ROUND OF REVIEW

Your manuscript, "Modern planktonic Foraminifera: migrating is not enough", has now been seen by our referees, and in the light of their advice I am delighted to say that we can in principle offer to publish it. First, however, we would like you to revise your paper to address the points made by the referees, and to make some editorial changes to your paper so that it is as brief as possible and complies with our Guide to Authors.

Authors: We would like to thank the editor and the anonymous reviewers for their helpful feedback and guidance throughout the review process. We are pleased with the positive response and have carefully addressed the remaining concerns raised by both Reviewers #1 and #2.

To address the Reviewer #2 comments, we have provided further clarification on the data, methods, and limitations in the text to enhance transparency. Additionally, we have included Figure R1 (from the Rebuttal letter) in the Supplementary Information, as it specifically responds to some of Reviewer #2 concerns. All revisions and additions to the manuscript have been highlighted in red for your convenience.

Referees' comments:

Reviewer #1 (Remarks to the Author):

This is my third review of the manuscript. This is a much improved and very much clearer manuscript. All my previous comments have been adequately addressed. The additional extended data figures are very welcome, and a great addition to the manuscript.

As previously stated, this is an extremely interesting and novel study, bringing together multiple data sets to determine recent and potential future plankton changes. It is widely accepted that anthropogenic warming will have an impact on life on Earth, but analysis and quantification of the effects that have already taken place over the last 50 or so years have been lacking. In this paper, the authors use FORCIS (a database of planktonic foraminifera with data going back to 1910). This manuscript builds on Chaabane et al. (2023) Scientific Data where the FORCIS database was published. The ability to examine how plankton communities have changed vertically, spatially and through time is particularly novel and will be of interest to a wide audience, including marine biologists and paleontologists interested in the macro and micro fauna and flora, and ocean and climate modellers.

Regarding Data and Methodology, and the validity of the approach, I still have one comment of confusion about the data handling and some minor edits where the text lacked some clarity.

Authors: We sincerely thank the Reviewer #1 for the helpful and positive feedback on our manuscript. The comments have been invaluable in enhancing our work.

Data and Methodology: I'm still unsure how some of the species/subspecies are treated in the dataset. For example it is *G. ruber ruber* on lines 139-140, 207, but only *G. ruber* line 143. Then line 343, *G. ruber* is separated into *G. ruber albus* and *G. ruber elongatus*, but with no mention of *G. ruber ruber*. This becomes very challenging when looking at northern displacement of occurrence (extended data Fig. 7) where *G. ruber* has a poleward migration, but *G. ruber ruber* has equatorward migration. I suggest that the authors lump and simply use *G. ruber* and explain in the text that this includes *G. ruber ruber*, *albus* and *elongatus*, and that the reporting in the data compilation prevents these species/subspecies being separated. This also has implications for the figures and tables. Alternatively, it needs to be stated in the text of how the distributions of *G. ruber*, differ from those of *G. ruber ruber*. Currently these species/subspecies have separate distributions in Fig. 2 and extended data fig. 8.

Reply: We appreciate the reviewer's insightful comments and understand the concern regarding the treatment of species in the dataset. The species *G. ruber* (white) was usually lumped to include both *G. ruber albus* and *G. elongatus*, as they were initially thought to occupy similar ecological niches. Thus, we clustered these two morphospecies and distinct them from that of *G. ruber ruber* (pink), due to the availability of morphospecies data over an extended time period and the ease of distinguishing certain morphospecies.

To avoid confusion, we clarified this distinction in the revised manuscript by adding a sentence to the Methods section.

The revised text now reads: **"In this study, several morphospecies were grouped (lumped) together for data analyses such as *G. ruber* (*G. ruber albus* and *G. elongatus*), *G. truncatulinoides* (*G. truncatulinoides* left and *G. truncatulinoides* right) and *T. sacculifer* (*T. sacculifer* no sac and *T. sacculifer* sac). However, *G. ruber ruber* (pink) was analyzed separately from *G. ruber* (white)." in lines 492-496.**

Minor edits

Line 52 abstract: the text states "this would radically alter low-latitude planktonic foraminifer communities", given that this is in the abstract, I think it should be more specific, in what way are the communities altered?

Reply: We have revised the abstract to clarify the specific impact on low-latitude PF communities. The sentence now reads: **"While these species may replace high-latitude ones through poleward shifts, this would reduce low-latitude Foraminifera diversity."** in lines 52-54.

Line 68: change "enhance the" to "alter". The increase in nutrient availability would depend on the position in the water column.

Reply: Done in line 68.

Line 117: change "higher" to "greater" to avoid the combination of higher and high.

Reply: Changed in line **117**.

Line 230: reference needed for *G. bulloides* environment.

Reply: In this section, we described observations stated from figure 4B and Extended data Fig. 9 and this goes in the same direction and perspective of what has been observed in culturing studies of *G. bulloides* under varying pH conditions such as the study of Sykes et al. (2024) in line **230**.

Sykes, F. E., Meilland, J., Westgård, A., Chalk, T. B., Chierici, M., Foster, G. L., & Ezat, M. M. Large-scale culturing of the subpolar foraminifera *Globigerina bulloides* reveals tolerance to a large range of environmental parameters associated with different life-strategies and an extended lifespan. *J. Plankton Res.* **46**, 403–420 (2024).

Line 412: change “This data was” to “These data were”

Reply: Changed in line **592**.

Line 450: “It was assessed” not clear what “it” is referring to.

Reply: Corrected. the “it” refers to “**The vertical range of PF**” in line **577**.

Sometimes pre-industrial, sometime preindustrial

Reply: Corrected.

Referee #2 (Remarks to the Author):

I had reviewed this paper previously and find it substantially improved, especially with regards to explanations in the methodology. My reservations remain about the comparability of the collection methods included here, though I appreciate the steps taken to discuss these issues in the response comments.

Authors: We would like to thank the reviewer #2 for the helpful comments and suggestions. We sincerely appreciate the constructive feedback provided.

I greatly appreciate the comparison provided between net and sediment trap data shared in the reviewer comments and find it heartening to see the similarity in diversity between the two. However, it is difficult to believe that this comparison wouldn't underestimate diversity from sediment traps. While the average collection period may only be 15 days, in practice few sediment traps are collecting only for 15 days in each location. Rather, many will be collecting for months or years. Thus, at a given location, a sediment trap will collect through a wider variety of seasons and conditions than a single tow at the same location, inevitably sampling more diversity.

Reply: We appreciate the reviewer's helpful comment. We acknowledge that sediment traps, which collect over an extended period, could indeed capture a broader range of seasonal and environmental conditions compared to the single-time “snapshot” provided by plankton net tows. In the FORCIS database, a sediment trap sample refers to the collection cup that gathers samples over an average period of 15 days. However, each mooring setup, which may be in place for months or years, provides a time-integrated profile.

For Figure 1, we compared plankton net profiles, which consist of multiple samples collected simultaneously from the same location using the same device, with sediment trap samples. The sediment trap samples in our database were collected over approximately 15 days and mostly from depths greater than 3000 m. Each sediment trap's entire time series is represented by a unique profile ID (Chaabane et al., 2023, Scientific Data).

In Figure 1, we observe that the diversity derived from plankton net profiles often exceeds that from sediment trap samples (yellow vs. orange dots). In some instances, such as in the Southern Hemisphere and northern mid-latitudes, sediment trap samples show slightly greater diversity than certain plankton tow samples. This discrepancy may be due to the limited temporal coverage of plankton nets, which cannot comprehensively capture all seasonal variations, while sediment traps, despite their time-averaging effects, integrate over a broader range of conditions. The greater diversity often seen with plankton nets can be attributed to their generally higher resolution and better sample preservation.

In the revised version, we clarify this in this: **“To address the potential disparity between sediment trap and plankton net data, sediment trap samples, which were collected over an average period of 15 days, were compared to plankton net profiles that consist of multiple samples gathered at the same location and time. The sediment trap's collection period would capture a representative range of species comparable to those found in plankton net profiles taken at similar locations (Supplementary Fig. 3).”** in lines 536-541.

This response also side steps the issue of the comparison to sediments (ForCens), in which case the comparison is a snapshot of species richness (plankton tows) compared to species richness in a sample aggregating over years, centuries, or longer. It should thus not be surprising that sediments would sample a greater diversity of species under most circumstances.

Reply: We appreciate the reviewer’s insightful comment regarding the comparison between plankton net data and sediment samples from ForCenS. We acknowledge that surface sediment samples, which aggregate data over extensive time periods, often capture a broader diversity of species compared to the snapshot nature of plankton tows. This difference means that sediments, due to their long-term accumulation, typically reflect higher species richness.

In our study, we focus on comparing the presence and absence of species rather than absolute species richness to circumvent these differences in time integration. Figure 1 illustrates our comparison of plankton net profiles, which involve multiple samples, with sediment trap samples collected every 15 days and surface sediment samples. Moreover, given that it takes 3 to 10 days for Foraminifera to sink from the surface to depths greater than 3000 m (Takahashi and Bé, 1984), we expect that a sediment trap sample with a 15-day collection period would capture most of the Foraminifera species present in a plankton net profile taken at the same location and time. Despite its time-averaging effect on the surface sediment samples, they should collect and reflect a similar community composition as the plankton net profiles and

the sediment traps samples during these 15 days of collection. Despite the inherent biases related to preservation, ontogeny, and seasonality, our results indicate that modern plankton net data generally reveal higher species richness in high-latitude regions compared to pre-industrial sediment data. This suggests that recent increases in species diversity are evident, despite the different temporal scales and sampling biases of the datasets.

We believe that the statement in the manuscript is cautious enough to illustrate the complexity of assessing biodiversity from those different datasets:

“Although both diversity estimates are influenced by biases such as preservation, ontogeny, and seasonality, our comparison shows that modern PF diversity is greater in high-latitude regions and lower near the Equator, relative to pre-industrial levels.” in lines 115-118.

Finally, the addition of Fig S2 is wonderfully informative. However, I find this potentially less reassuring than the authors do. Fig S2 highlights not only substantial trends in the types of sampling available through time, but also the relatively low richness captured by CPR.

Reply: The CPR sample locations presented in figure 1 and Supplementary Fig. 2 explain the low diversity coming from the Southern Ocean. The CPR device deployed in the Southern Ocean captured up to five species simultaneously, with an average of one to three species. This lower species richness reflects the naturally low diversity in this region.

Minor comments:

54 – quotations not needed

Reply: Quotations removed

159 – *N. dutertrei* has photosymbionts (Takagi et al., 2019 as cited, but also Bird et al., 2017 and others)

Reply: We agree that the paper of Takagi et al., 2019 was showing that *N. dutertrei* was facultatively hosting photosymbionts of different groups, in agreement with Bird’s study on a limited set of Foraminifera from the Eastern Pacific coast. Therefore, we only mention *N. incompta* and *G. inflata* which are unequivocally barren of photo-symbionts. This sentence was revised as follows: **“In the mid-latitudes, symbiont-barren species such as *G. inflata* (40±5 m from ~30 to 70 m), and *N. incompta* (40±4 from ~10 to 50 m) significantly descended in the thermocline and mixed layer (Fig. 2B).”** in lines 158-161.

211 – As pointed out previously, this model specifically takes into account a non-spinose and symbiont bearing species. This is potentially quite significant in a calcification context as the presence of symbiosis should alter the immediate carbonate microenvironment regardless of spine presence. Moreover, both the species mentioned are non-spinose, so by the logic of this sentence, shouldn’t they also be impacted by ocean acidification rather than advantaged by their size? I wonder if something like Henehan et al., 2017 showing lower CI in smaller individuals might not better bolster the claim being made here.

Reply: Thank you for raising this point. We agree that the calcification intensity/cost is also implicitly covered by Henehan et al. (2017) which is supporting our case of smaller vs. larger

Foraminifera. We have replaced the reference of Grigoratou et al. (2019) by the paper of Henehan et al. (2017) in the respective place (section line 208-215):

“Two different concurrent processes may explain this: first, the change in calcification intensity due to fluctuations in carbonate chemistry could provide an advantage to smaller species such as *G. uvula* and *G. glutinata*³⁹; second, this expansion could be attributed to the wide temperature tolerance of *G. uvula*⁴⁰⁻⁴¹ (Extended Data Figs. 4 and 5).”

Henehan, M. J., Evans, D., Shankle, M., Burke, J. E., Foster, G. L., Anagnostou, E., Chalk, T. B., Stewart, J. A., Alt, C. H. S., Durrant, J., and Hull, P. M.: Size-dependent response of foraminiferal calcification to seawater carbonate chemistry, *Biogeosciences*, 14, 3287–3308, <https://doi.org/10.5194/bg-14-3287-2017>, 2017.

329-332: Thank you for the additional detail here. Could you please provide or cite the equations used?

Reply: In the revised version we added the following equations:

“We were able to accurately quantify the abundance of coarse fractions down to 100 μm , by applying these following equations:

$$C_{sz_norm}^{\infty} = C_{sz_inf}^{sz_sup} \frac{f_{max} - f_{sz_norm}}{f_{sz_sup} - f_{sz_inf}}$$

where sz_inf and sz_sup denote the lower and upper size limits of the measured size fraction, respectively; sz_norm represents the normalization size; and f_{S_z} is the multiplication factor associated with S_z , calculated as follows:

$$f_{S_z} = 1 + (f_{max} - 1) \frac{(S_{sup,k} - S_{sup,1})}{(S_{sup,k} - S_{sup,1}) + (S_{half} - S_{sup,1})}$$

with $S_{sup,1}$ and $S_{sup,k}$ are the upper size limits of size class 1 and k, respectively. While S_{half} and f_{max} are reported in Chaabane et al.⁶⁰.” in lines 473-482.”